



# The influence of snow on sea ice as assessed from simulations of CESM2

**Marika M. Holland**[1]**, David Clemens-Sewall**[2]**, Laura Landrum**[1]**, Bonnie Light**[3]**, Donald Perovich**[2]**,**
**Chris Polashenski**[2]**, Madison Smith**[3]**, and Melinda Webster**[4]

[1]Climate and Global Dynamics Laboratory, National Center for Atmospheric Research, Boulder, CO, USA
[2]Thayer School of Engineering, Dartmouth College, Hanover, NH, USA
[3]Polar Science Center, Applied Physics Laboratory, University of Washington, Seattle, WA, USA
[4]Geophysical Institute, University of Alaska Fairbanks, Fairbanks, AK, USA

**Correspondence:** Marika M. Holland (mholland@ucar.edu)

**Abstract.** We assess the influence of snow on sea ice in experiments using the Community Earth System Model version 2 for a preindustrial and a 2xCO2 climate state. In the preindustrial climate, we find that increasing simulated snow accumulation on sea ice results in thicker sea ice and a cooler climate in both hemispheres. The sea ice mass budget response differs fundamentally between the two hemispheres. In the Arctic, increasing snow results in a decrease in both congelation sea ice growth and surface sea ice melt due to the snow's impact on conductive heat transfer and albedo, respectively. These factors dominate in regions of perennial ice but have a smaller influence in seasonal ice areas. Overall, the mass budget changes lead to a reduced amplitude in the annual cycle of ice thickness. In the Antarctic, with increasing snow, ice growth increases due to snow–ice formation and is balanced by larger basal ice melt, which primarily occurs in regions of seasonal ice. In a warmer 2xCO2 climate, the Arctic sea ice sensitivity to snow depth is small and reduced relative to that of the preindustrial climate. In contrast, in the Antarctic, the sensitivity to snow on sea ice in the 2xCO2 climate is qualitatively similar to the sensitivity in the preindustrial climate. These results underscore the importance of accurately representing snow accumulation on sea ice in coupled Earth system models due to its impact on a number of competing processes and feedbacks that affect the melt and growth of sea ice.

## 1 Introduction

Snow is the most reflective natural material on Earth, and its presence plays an important role in the cooling of the globe (e.g., Perket et al., 2014). It is also a very effective thermal insulator (e.g., Sturm et al., 2002a) and thereby reduces the transfer of heat from the underlying surface (e.g., soil or sea ice) into the atmosphere. Its high albedo and its role in heat transfer give it a prominent role in Earth's climate system.

Snow on sea ice is influenced by numerous processes that differ spatially and over time (see Webster et al., 2018, and Sturm and Massom, 2017, for recent reviews). Due to snow optical and thermal properties, it can have competing influences on the thermodynamic ice mass budgets that differ by season. During the ice growth season, when sunlight is reduced or nonexistent, the insulating effect of snow dominates. Although the snow thermal conductivity varies with snow density and metamorphic properties (e.g., Sturm et al., 1997, 2002a; Colonne et al., 2011), field measurements indicate that it is roughly an order of magnitude smaller than that of sea ice. Because of this, a thicker snow cover reduces the amount of heat conducted through the ice pack, resulting in lower surface heat loss to the atmosphere and reduced sea ice growth. In contrast, during the sunlit season, the presence of snow leads to a sizable increase in the surface albedo and less absorption of solar radiation in the snow, ice, and underlying ocean. This reduces ice melt. The amount of snow on the ice pack can also drive snow–ice formation, which occurs when the weight of the snow submerges the ice–snow interface below the water line, causing seawater to flood the snow and, in

freezing conditions, form ice. This ice growth mechanism is significant in the Antarctic (e.g., Maksym and Jeffries, 2000; Massom et al., 2001). Although it plays a considerably small role than in the Antarctic, snow–ice formation has also been observed in the Arctic (e.g., Granskog et al., 2017), with the potential being particularly high in the Atlantic sector of the Arctic (e.g., Merkouriadi et al., 2020).

Considerable work has elucidated the role of snow in the evolution of the surface albedo of sea ice. As discussed by Perovich et al. (2002), the albedo of Arctic sea ice undergoes a series of phases in its seasonal evolution, in which the presence and state (dry or melting) of snow cover play a central role. When dry snow is present in the spring, the albedo is high (0.8–0.9) and spatially uniform. As melt is initiated, sometimes aided by episodic events such as rainfall on the snow cover, snow grains increase in size, leading to a darkening of the surface (to an albedo of around 0.7). With further melt, ponding and bare ice conditions occur, darkening the surface further and contributing to high spatial heterogeneity in surface albedo. In the Antarctic, the ice is mostly snow covered with an absence of ponds, even in summer, and thus the surface albedo remains high and undergoes a much smaller annual cycle than that in the Arctic (e.g., Andreas and Ackley, 1982; Massom et al., 2001; Brandt et al., 2005).

Changes in the timing of Arctic seasonal transitions in a warming climate act to reduce the annual average surface albedo. Satellite data suggest trends of earlier melt onset and a lengthened melt season since 1979 (Stroeve et al., 2014). Satellite observations also indicate reductions in the Arctic surface albedo since the 1980s in part due to reductions in snow on sea ice (Zhang et al., 2019). Climate model simulations (Holland and Landrum, 2015) suggest that changes in the Arctic Ocean surface properties, including an earlier snow melt onset and lengthening snow-free season, remain important for albedo reductions into the future. Additionally, studies indicate that Arctic winter ice growth can increase in a warming climate as a direct consequence of the reduced insulating effect of a thinning ice cover with reduced snow (e.g., Bitz and Roe, 2004; Stroeve et al, 2018; Petty et al., 2018). Thus previous work suggests that for the evolving Arctic thermodynamic sea ice mass budgets, reduced snow has competing influences by reducing the albedo, thereby increasing summer melt, and by increasing conduction of heat through the ice, thereby increasing winter growth.

In the Antarctic, other important processes can play a role. Because of high-snowfall and thin-ice conditions, surface flooding is prevalent and causes considerable snow loss and ice gain through snow–ice formation (e.g., Maksym and Jeffries, 2000; Massom et al., 2001). With high winds and less concentrated ice relative to the Arctic, windblown snow loss to leads is also important for the snow mass budgets (Leonard and Maksym, 2011). The presence of high ocean heat content causes Antarctic sea ice to primarily melt from below (e.g., Gordon, 1981). This allows the ice to remain largely snow covered even in summer (e.g., Massom et al., 2001; Brandt et al., 2005). Consequently, there is a lack of melt ponding on Antarctic sea ice (Andreas and Ackley, 1982).

Models have been used to assess the integrated role of snow and its thermal and optical properties on sea ice and climate conditions. Single-column model studies of Arctic sea ice (Maykut and Untersteiner, 1971; Semtner, 1976) indicate that ice thickness initially decreases with increasing snow depths due to the insulating effect of snow but then increases greatly due to the albedo effect. Under very thick snow conditions, these studies predict a lack of snow-free summers with no surface sea ice melt. A study with a simple energy balance climate model (Ledley, 1991) suggested that the overall impact of snow on sea ice is to cool the climate, with the albedo effect of snow dominating. Simulations with an ice–atmosphere coupled model (Wu et al., 1999) highlighted the importance of snow–ice formation for the sensitivity of Antarctic sea ice and also showed that the Arctic and Antarctic respond differently to increasing snow amounts. Sensitivity studies in sea ice (e.g., Powell et al., 2005), ice–ocean coupled (e.g., Fichefet and Morales-Marqueda, 1999), and fully coupled (e.g., Blazey et al., 2013) model frameworks indicate that the sea ice mass budgets and other climate characteristics are affected by the snow thermal properties. However, limited recent work has been done to investigate the overall impacts of changing snow conditions in coupled climate models with active atmospheric feedbacks.

There are multiple reasons why snow accumulation on sea ice may not be accurately represented in models. Significant uncertainties remain in our understanding of the physical processes driving precipitation in the polar regions, resulting in the differences between different reanalysis estimates of snowfall on Arctic sea ice being of the same order of magnitude as the amount of snowfall (Boisvert et al., 2018). Coupled climate models also exhibit considerable differences in Arctic precipitation (e.g., Kattsov et al., 2007), indicating high structural model uncertainty. Even if the modeled precipitation amounts are accurate, other processes modify the amount of snow on sea ice. Wind-driven blowing snow loss into leads could reduce snow on sea ice in the Antarctic by 50 % (Leonard and Maksym, 2011) and is typically not represented in climate models. This suggests that models are missing a potentially important Antarctic snow sink and may overestimate snow–ice formation. However, the net impact on sea ice mass budgets is unclear since much of the snow lost to leads may result in ocean supercooling and ice growth. Finally, sub-grid-scale processes could change the impact of snow on sea ice in ways that are analogous to having more or less snow. For example, Sturm et al. (2002a) found that accounting for the spatial heterogeneity of snow depth increased the regional effective thermal conductivity of the snow cover during the Surface Heat Budget of the Arctic Ocean (SHEBA) field campaign by at least 40 % relative to point measurements.

A better understanding of the influence of snow on sea ice for climate characteristics is important given the large

changes underway in the climate system. Indeed, observations suggest that snow on Arctic sea ice is declining in the warming climate in large part because of the later fall ice freeze-up and resulting loss of a sea ice platform to accumulate snow (Webster et al., 2014). Climate simulations project that these changes will continue into the future (Hezel et al., 2012; Webster et al., 2021), with implications for changing albedo and thermal ice growth feedbacks.

Here we address a very basic question: does changing the amount of snow on sea ice lead to reductions or increases in sea ice area and volume? We use experiments with a coupled climate model in which we impose changes in the amount of snowfall over the sea ice pack, effectively changing the amount of snow on sea ice in both hemispheres. This allows us to isolate the influence of snow on sea ice, to assess its regional and hemispheric dependence, and to determine the net impacts on the sea ice cover and polar climates while accounting for atmospheric feedbacks.

## 2   Methods

To investigate the role of snow on sea ice, we perform simulations using the Community Earth System Model 2 (CESM2; Danabasoglu et al., 2019). The atmosphere is modeled using the Community Atmosphere Model 6 (CAM6; Gettelman et al., 2019), and land is modeled using the Community Land Model 5 (CLM5; Lawrence et al., 2019). Sea ice is modeled using the CICE sea ice model version 5.1.2 (Hunke et al., 2015). This model includes an elastic–viscous–plastic (EVP) rheology (Hunke and Dukowicz, 2002), a sub-grid-scale ice thickness distribution (Holland et al., 2006), a multiple-scattering parameterization for solar radiation (Briegleb and Light, 2007) which allows for the presence of black carbon and dust aerosols (Holland et al., 2012) and the radiative effect of ponds (Hunke et al., 2013), and the mushy-layer thermodynamics of Turner and Hunke (2015) which includes prognostic salinity. Ice motion is obtained from the momentum equation (e.g., Hibler, 1979) and accounts for the influence of wind and ocean stresses, the Coriolis effect, sea surface slope, and internal ice stress. The model includes mechanical redistribution (Rothrock, 1975; Thorndike et al., 1975) based on the simulated sea ice convergence and shear, with ridging causing thin ice within the five-category sub-grid-scale ice thickness distribution to be converted into a smaller area of thicker ice while conserving the ice volume and internal energy.

Over the annual cycle, snow is accumulated through snowfall as determined by the atmospheric model. Snow is lost through melting, sublimation, snow–ice formation, and ridging. Melting is determined from a balance of fluxes at the surface. Snow–ice is parameterized to form when the weight of the snow depresses the snow–ice interface below sea level. A volume of snow is then flooded with seawater and consolidated into ice with an appropriate porosity and bulk salinity (Turner and Hunke, 2015). The amount of snow–ice formed is sufficient to raise the snow–ice interface to sea level. Snow is also lost to leads during ridging, with 50 % of the snow that is present on the ice participating in ridging being lost to the ocean. Regarding the albedo treatment of the snow, a snow grain radius is prescribed in the model for use in the radiative transfer (Briegleb and Light, 2007). The snow grains transition from a dry snow grain radius value to a melting snow grain radius value when the snow is close to the melting temperature. Rainfall has no direct impact on the simulated optical or thermal properties of the snow cover. The model uses a constant density ($330 \, \mathrm{kg \, m^{-3}}$) and thermal conductivity ($0.30 \, \mathrm{W \, m^{-1}}$ per degree) for snow and does not include any windblown snow redistribution or loss to leads. The model parameterizations are identical in both hemispheres. More details on the sea ice model configuration within CESM2 are available in Bailey et al. (2020).

The CESM2 has a reasonable simulation of Arctic and Antarctic sea ice, including both the mean state and variability (e.g., DeRepentigny et al., 2020; DuVivier et al., 2020; Singh et al., 2020; Raphael et al., 2020). However, simulated Arctic ice is thin compared to observations over the historical period (DuVivier et al., 2020). Simulated snow on Arctic sea ice is also thin compared to observations and accumulates too slowly in the fall (Webster et al., 2021).

For this study, we perform sets of preindustrial and doubled $CO_2$ (2xCO2) simulations with modifications to the amount of snowfall over sea ice. The model uses the full atmosphere, sea ice, and land models of CESM2 but a simplified ocean. In particular, a slab ocean model (SOM) configuration replaces the full depth dynamic ocean model of the standard CESM2. This SOM model configuration has typically been used to assess equilibrium climate sensitivity (e.g., Bacmeister et al., 2020). The SOM computes a single surface ocean averaged temperature that is representative of a fixed-depth mixed layer. This evolves subject to time-evolving surface heat fluxes, a prescribed ocean heat flux convergence associated with lateral and vertical ocean dynamics, and a prescribed ocean mixed layer depth that varies spatially. The prescribed ocean heat flux convergence and mixed layer depth are obtained from the fully coupled CESM2 preindustrial control run (Danabasoglu et al., 2020), and the heat convergence accounts for the net dynamic ocean heat flux associated with advection and mixing (Bitz et al., 2012). The use of a SOM configuration allows the system to reach an equilibrated state rapidly (in about 30 years versus hundreds of years for the standard model) and allows us to perform multiple sensitivity simulations. Our simulations are run for 50 years, and we define the climatology as the average of the last 20 years of simulation after the system has mostly equilibrated and large-scale drift has declined. For example, the trend in hemispheric ice volume for the years 31–50 which are analyzed here is less than 2 % of the mean. Note that the use of the SOM means that any ocean dynamical feedbacks are excluded and so changes in

**Table 1.** Experiments and annual mean snow volume in the Northern Hemisphere (NH) and Southern Hemisphere (SH) relative to the control experiment.

| $F_{snow}$ value | Annual mean snow volume in the experiments relative to the control ($F_{snow} = 1.0$) experiment in the preindustrial climate | |
| --- | --- | --- |
| | NH | SH |
| 0.0 | 0.00 | 0.0 |
| 0.25 | 0.21 | 0.20 |
| 0.5 | 0.46 | 0.51 |
| 0.75 | 0.72 | 0.78 |
| 1.00 | 1.00 | 1.00 |
| 1.25 | 1.27 | 1.29 |
| 1.5 | 1.66 | 1.77 |
| 1.75 | 1.94 | 2.09 |

ice growth and related ice–ocean salt and freshwater fluxes across our sensitivity simulations will not affect ocean vertical mixing or dynamic ocean heat transport. The SOM does allow the ocean surface temperature to respond to evolving surface heat fluxes, including changes in shortwave absorption associated with transmission through sea ice and through open-water areas. SOM integrations with 2xCO2 levels provide an estimate of the equilibrated response of the system which is considerably higher than the transient climate response (e.g., Bitz et al., 2012).

In the standard model, the atmospheric component computes a snowfall amount based on the environmental conditions, including temperature, humidity, and cloud and aerosol properties (e.g., Gettelman and Morrison, 2015). In order to modify the snow amount on sea ice, we perform experiments in which the snowfall that the sea ice component receives from the atmosphere is multiplied by a constant factor ($F_{snow}$; Table 1). In doing this, snowfall on sea ice is increased (if $F_{snow} > 1$) or decreased (if $F_{snow} < 1$) and is gained or lost from the coupled system. We complete simulations with snowfall on sea ice completely removed ($F_{snow} = 0$) to nearly doubled ($F_{snow} = 1.75$). A value of $F_{snow} = 1$ is the standard model run in which no change is made to the snow received from the atmosphere model. Note that in the context of the global coupled model this method is not conservative in that we are adding or removing snow from the system by altering snowfall as it is received on the ice surface. However, because the changes we impose only affect snowfall over sea ice, they are quite small relative to precipitation on a global mean.

In these experiments, the atmosphere will respond to the changing sea ice state, and so atmospheric feedbacks are active. This will include the influence of changing surface heat, moisture, and momentum fluxes on air temperature, humidity, winds, and cloud properties, among other conditions. As a consequence, the exact amount of snowfall is not specified

in our experimental design and can vary with the climate. This, along with differences in the fall ice area platform upon which snow can accumulate, results in relative snow volume amounts that vary somewhat from the $F_{snow}$ multiplication factor (Table 1). However, the use of the $F_{snow}$ factor does effectively modify the amount of snow on sea ice without directly impacting other aspects of the climate system. This allows us to isolate the role of snow on sea ice and assess the net impacts on the sea ice and climate state.

## 3 Results

### 3.1 Control sea ice climate

Some aspects of the sea ice climatology from the control ($F_{snow} = 1.0$) simulation are shown in Fig. 1. The annual cycle of ice area is compared to present-day observations (Fetterer et al., 2017) in both hemispheres to provide a rough estimate of simulation quality given that observed ice conditions are not available for the preindustrial time period of the model simulations. In the Arctic, the simulated ice area (Fig. 1c) is larger throughout the annual cycle, consistent with the lower greenhouse gas levels present in the simulated preindustrial control climate, as well as a lack of significant anthropogenic ice loss that has been observed in recent decades (e.g., Mueller et al., 2018). The annual cycle of the Antarctic sea ice area (Fig. 1f) agrees very well with present-day observations, which is consistent with a much smaller ice loss observed in recent decades in the Southern Hemisphere (e.g., Parkinson and Cavalieri, 2012). Although ice thickness observations are limited, especially in the Antarctic (e.g., Meredith et al., 2021), the simulated spatial pattern and mean values of ice thickness (Fig. 1a and d) are generally consistent with observations or observationally constrained model simulations (e.g., Worby et al., 2008; Schweiger et al., 2011; Labe et al., 2018). As observed, the Antarctic ice pack is considerably thinner than that in the Arctic. The monthly maximum in snow depth occurs in May for the Arctic (consistent with observations, Warren et al., 1999) and October for the Antarctic. In the Arctic, the simulated May snow (Fig. 1b) is thickest near Fram Strait and thinner within the shelf regions. Although the simulation is for a preindustrial period, this general structure is in agreement with recent observations (e.g., Webster et al., 2018; Fig. S1 in the Supplement). In the Antarctic, the simulated October snow depth pattern (Fig. 1e) closely matches that of the ice thickness, which is in part because thicker ice is less prone to snow–ice formation and so can maintain a thicker snowpack. While observations are limited in the Antarctic, the model appears reasonable (e.g., Webster et al., 2018; Kacimi and Kwok, 2020; Fig. S2). Note that the use of a constant snow density in the model could lead to snow thickness biases relative to observations even if the snow mass is well simulated. The prescribed value of $330\,\mathrm{kg\,m^{-3}}$ is representative of winter

measurements in the Arctic (Warren et al., 1999; Sturm et al., 2002a) and Antarctic (Massom et al., 2001), but seasonal and regional deviations are likely. For example, in the Arctic, the prescribed model density is high in the autumn relative to observations (Blazey et al., 2013). Regional weather and snow processes will also drive spatial variations in density. While there are too few observations available to characterize regional climatological snow density, we would expect that regions with widespread flooding (i.e., Bellingshausen, Amundsen, and Ross seas) and frequent melt–freeze cycles (southerly latitudes in the Arctic and northerly latitudes in the Antarctic) likely have higher snow densities. In contrast, regions with calm, dry, cold conditions (i.e., autumn in the northeastern Beaufort Sea) will tend to have lower snow densities.

The simulated sea ice mass budget that drives the annual cycle is considerably different between the two hemispheres (Fig. 2). The model simulates several distinct ice growth and melt processes. Ice accumulation processes include congelation ice growth onto the base of an existing sea ice pack, frazil ice formation associated with the supercooling of ocean water, and snow–ice formation. Melting processes are comprised of surface melting at the top of the ice, basal melting at the bottom of the ice, and lateral melting at the sides of floes. Dynamic processes do not directly act as a gain or loss of ice mass but do transport sea ice regionally and drive ice ridging, which causes conversion of thinner ice to thick ice with a smaller areal coverage while conserving ice volume. There are model limitations that will affect the simulated mass budgets. The model includes a somewhat simple lateral melt parameterization that assumes a constant floe size (Steele, 1992), and ice–wave interactions are not included. The frazil ice parameterization (Hunke et al., 2015) produces sea ice based on the supercooling of the ocean but does not account for factors such as windblown dynamics in leads. These simplifications affect the simulated ice mass budgets, typically leading to low lateral melting (Roach et al., 2018) and low frazil ice formation (Wilchinsky et al., 2015) when compared to more advanced parameterizations.

In the Northern Hemisphere (Fig. 2a), ice is primarily accumulated through congelation growth from October through April. Frazil ice formation plays a small role. This ice growth is balanced by melting, which occurs year-round with winter basal melt occurring along the ice edge in the North Atlantic and North Pacific. Melt is largest from June through August when the perennial ice-covered region of the Arctic basin experiences sizable melting at both the surface and the base of the ice. The surface ice melt becomes sizable in June as the snow-free area of the sea ice increases. Other terms in the ice mass budget, including snow–ice formation and lateral melting, only contribute slightly to the Arctic mean budgets.

The situation is considerably different for the thinner ice pack of the Southern Hemisphere (Fig. 2b). From March through November, ice is accumulated through a combination of congelation ice growth, which dominates at the begin-

ning of the growth season, and snow–ice formation, which dominates at the end of the season. This seasonal transition is in part associated with the region of the ice pack that experiences the majority of growth. This shifts from primarily coastal congelation growth in the early part of the season to snow–ice formation in the thinner, seasonal ice zone later in the season. The substantial snow–ice formation in the Antarctic is associated with the thin ice pack and a snowfall which is over 3 times greater than that in the Arctic. Frazil ice formation also contributes a small amount of ice mass, particularly in coastal regions (as shown by Singh et al., 2020). Observations suggest that a higher fraction of frazil ice is present in Antarctic sea ice than that simulated here (e.g., Lange and Eicken, 1991; Worby et al., 1998). This may be related to limitations in the parameterization of frazil ice formation (Wilchinsky et al., 2015). In the model, ice melt happens year-round in the Antarctic but is largest from September through February. This melt is almost entirely due to melting at the base of the ice pack. In contrast to the Arctic, surface ice melt is negligible in all months. The differences in basal melting are associated with different oceanography in the two regions, which allows for an annual mean ice–ocean heat exchange in the Antarctic that is about 3 times larger than that in the Arctic.

These differences between the sea ice mass budgets in the two hemispheres generally agree with observed mass budget information. In particular, the lack of surface melting in the Antarctic is consistent with observations that indicate an ice cover that remains largely snow covered, even in summer (Massom et al., 2001; Brandt et al., 2005), with few surface melt ponds (Andreas and Ackley, 1982). It is also consistent with heat budget considerations that suggest most Antarctic ice melts from below (Gordon, 1981).

## 3.2 Influence of different snow depths on the sea ice and atmosphere state

By adjusting the snowfall received by the sea ice from the atmosphere, we effectively change the hemispheric snow volume and the local snow depth on the sea ice cover. While we use a constant multiplication factor to make this modification, this does not necessarily result in an equivalent factor change in the hemispheric snow volume as discussed in Sect. 2 (Table 1). This is in part because the mean ice area changes in the sensitivity simulations, and this affects whether and when there is a platform on which snow can accumulate (e.g., Hezel et al., 2012; Webster et al., 2014). Additionally, as discussed further below, atmospheric conditions are responsive to the changing ice climate affecting air temperature and snowfall amounts. While these feedbacks are active in our simulations, our modifications to the model do result in a wide range of snow volume amounts on the sea ice across the different simulations, allowing us to assess the net effect of changing snow amounts on the sea ice state.

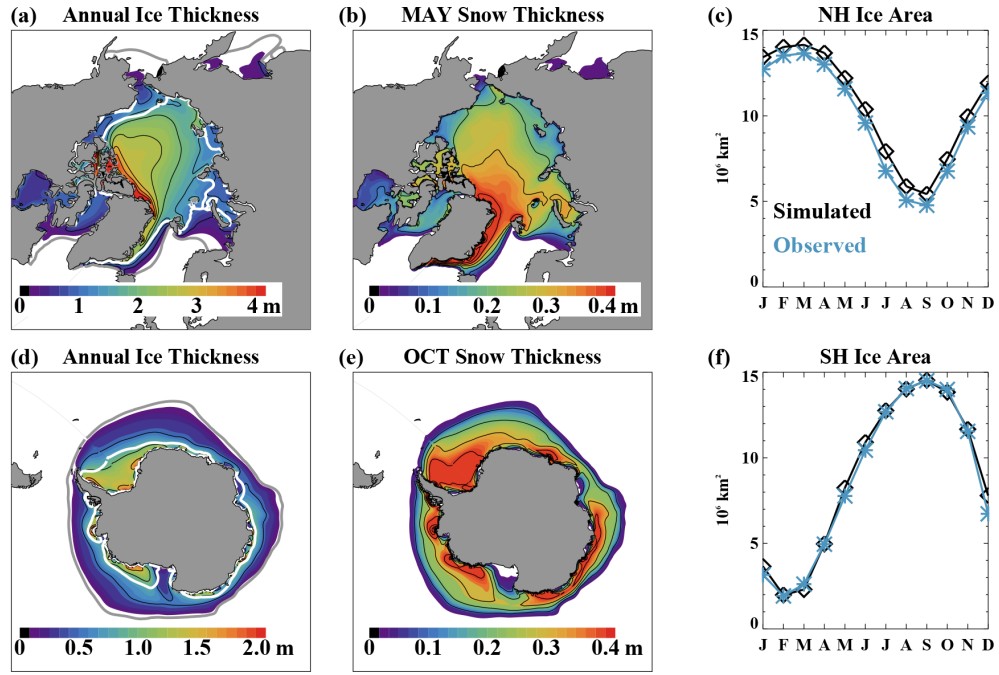

**Figure 1.** The simulated Northern Hemisphere **(a)** annual ice thickness, **(b)** May snow depth, and **(c)** ice area as well as Southern Hemisphere **(d)** annual ice thickness, **(e)** October snow depth, and **(f)** ice area. On the ice thickness plots, the contour interval for the black lines is 0.5 m, and the simulated 15 % ice concentration for March and September is shown in the bold white and gray contour lines. For the snow thickness plots, the contour interval for the black lines is 0.1 m. The observed ice area averaged from 1979–2013 shown on panels **(c)** and **(f)** is from the NSIDC sea ice index (Fetterer et al., 2017).

Figure 3 shows the mean annual cycle of Northern Hemisphere snow volume, ice volume, and ice area. The differences in the annual mean snow volume across the runs are quite similar to the $F_{snow}$ values applied to the snow-fall (Table 1) but somewhat larger in the $F_{snow} = 1.5$ and $F_{snow} = 1.75$ cases. In these high-snowfall cases, the annual mean snow volume is 1.66 (for $F_{snow} = 1.5$) and 1.94 (for $F_{snow} = 1.75$) times greater than that in the control run. This is associated with larger areas of the Arctic which never reach snow-free conditions in summer and high ice concentrations in fall which provide a platform for snow accumulation. The ice volume and ice area generally increase with increasing snow. The simulation with zero snowfall has a fundamentally different climatology with very thin ice and near ice-free conditions in August–October. The high-snowfall cases ($F_{snow} = 1.5$ and $F_{snow} = 1.75$) retain summer snow in the hemispheric values. Thus, for the Arctic, the simulations can be roughly categorized as following into three regimes that include no snow, seasonal snow, and perennial snow cases. Although the snow volume annual cycle is larger with higher snow amounts, the amplitude of the ice volume annual cycle declines, and ice volume differences across the runs are largest during summer.

The nonlinearity of the Northern Hemisphere sea ice response to changing snow depth is apparent in the summary results shown in Fig. 4. The annual mean ice volume and minimum ice area are very low in the case of zero snow depth. There is a significant jump in ice volume and area from the $F_{snow} = 0$ to the $F_{snow} = 0.25$ case, and the annual mean ice volume is then quite similar, ranging from $1.9 \times 10^{13}$ to $2.3 \times 10^{13}$ km$^3$ TS1, for maximum snow volume values ranging from $0.08 \times 10^{13}$ to $0.41 \times 10^{13}$ km$^3$ in simulations with $F_{snow}$ values from 0.25 to 1.25 (the seasonal snow cover cases). However, while the annual mean ice volume is similar across this range of values, the amplitude of the ice volume annual cycle does differ and consistently declines with increasing snow (Fig. 4b). This is largely caused by differences in the summer ice volume, which appears more sensitive than the winter ice volume amounts for these intermediate simulations (Fig. 3b). The ice area annual cycle also shows a reduction in the amplitude with increasing snow volume, with a large sensitivity in summer and smaller sensitivity in winter (Fig. 3c and from a comparison of the different $y$-axis range in Fig. 4c and d). This is consistent with the strong correlation between ice thickness and summer ice area (Blanchard-Wrigglesworth et al., 2011). The two highest $F_{snow}$ runs with values of 1.5 and 1.75 (perennial snow cover cases) have a maximum Arctic average snow volume of $0.5 \times 10^{13}$ km$^3$ TS2 or greater and exhibit significantly increased ice volume compared to the other simulations with a reduced annual cycle in ice area.

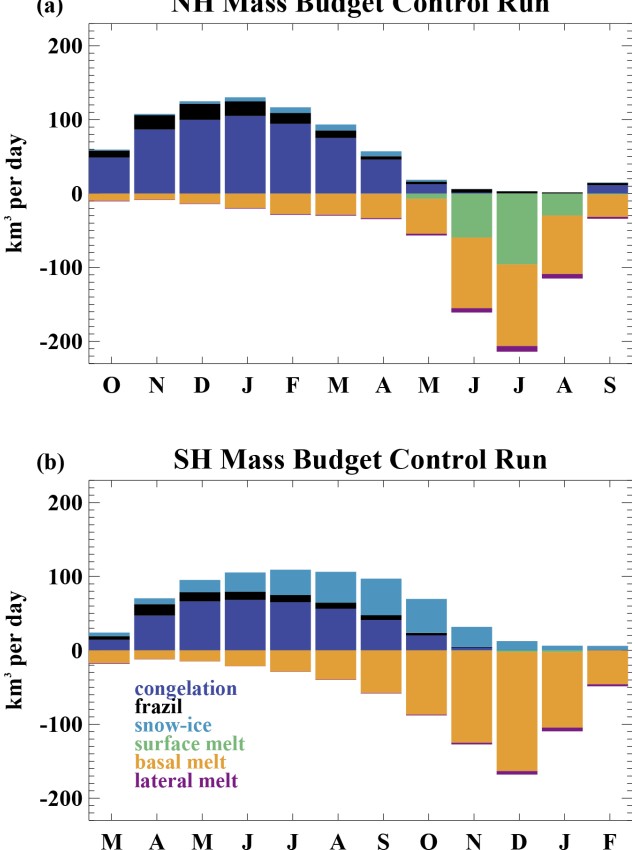

**Figure 2.** Annual cycle of the climatological ice volume budgets in the $F_{snow} = 1$ preindustrial control run for the **(a)** Northern Hemisphere and **(b)** Southern Hemisphere. The volume budgets can be directly converted into mass budgets using the constant sea ice density of $917\,\mathrm{kg\,m}^{-3}$ used in the model. Note that the annual cycle is shifted in both hemispheres to begin with the ice growth season (starting in October for panel **a** and March for panel **b**). Values are computed over the entire hemisphere.

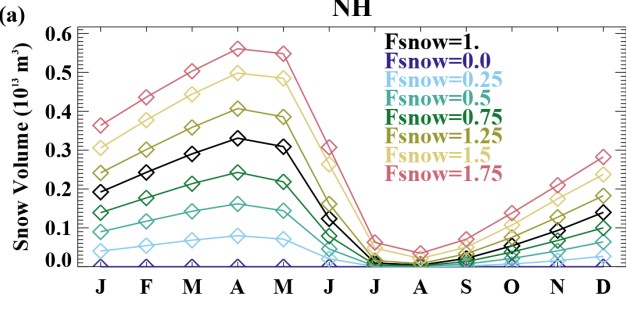

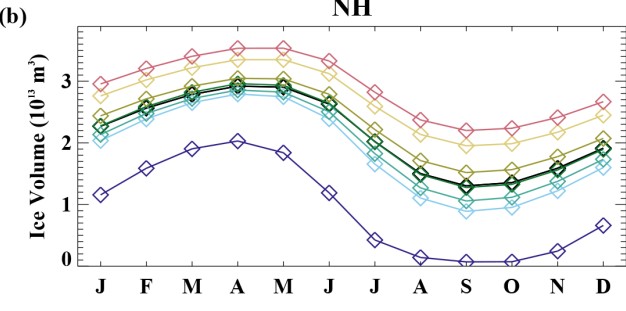

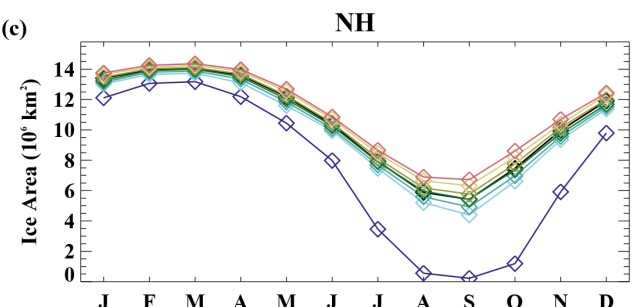

**Figure 3.** The Northern Hemisphere **(a)** snow volume, **(b)** ice volume, and **(c)** ice area in the different $F_{snow}$ experiments.

In the Southern Hemisphere (Fig. 5), the maximum snow volume is typically larger than that in the Northern Hemisphere. As in the Northern Hemisphere, the Antarctic ice volume and ice area increase with increasing snow volume (Figs. 5 and 6), although with a different nonlinear character. The amplitude of the ice volume annual cycle (Fig. 6b) increases with snow volume. This is in contrast to the Northern Hemisphere, where a decrease in the ice volume annual cycle amplitude occurs. In the no-snow case, ice-free conditions exist for about 4 months (Fig. 5c), and the timing of the maximum ice volume (Fig. 5b) occurs a month earlier than in the other simulations.

The differences in the ice climate associated with changing snow amounts influence and are influenced by changes in the polar atmospheric state. To analyze this, we assess mean atmospheric conditions for 70–90° N and 60–80° S, regions that are relevant to the underlying sea ice (Fig. 7). With in-

creasing snow on sea ice and thicker and more extensive ice in both hemispheres, the atmosphere is colder in both the Arctic and the Antarctic (Fig. 7a and d). This will in turn influence the sea ice–atmosphere heat exchange and further reinforce the ice state changes. The total high-latitude precipitation declines with the colder atmosphere (Fig. 7b and e), consistent with a reduced atmospheric moisture holding capacity and less evaporation from a more-ice-covered surface (e.g., Bintanja et al., 2014; Vihma et al., 2015). A greater fraction of this precipitation falls as snow in the colder climates. In both hemispheres, this generally offsets the overall decline in precipitation, resulting in only modest snowfall changes for the 70–90° N and 60–80° S annual means across the simulations (Fig. 7c and f). The exception is for the no-snow case in the 60–80° S region, where low snowfall occurs as the influence of temperature on precipitation phase dominates the snowfall response.

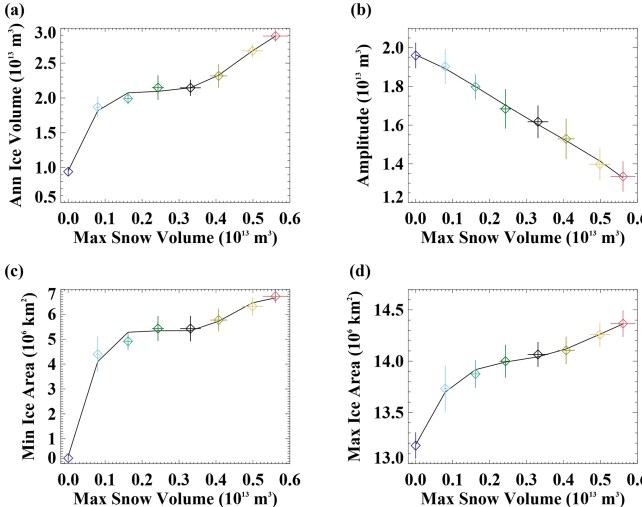

**Figure 4.** The relationship of the climatological Northern Hemisphere **(a)** annual ice volume, **(b)** amplitude of the ice volume annual cycle, **(c)** annual minimum ice area, and **(d)** annual maximum ice area versus the annual maximum snow volume. The diamonds show the simulated mean values, and whiskers are ± 1 standard deviation across the 20 years analyzed for each simulation. The colors are the same as in other figures. The black line is a fourth-order polynomial fit.

## 3.3 Differences in the ice mass budgets

Equilibrium sea ice conditions are obtained in the simulations when the long-term average annual ice melt is equal to the long-term average annual ice growth. Note that because of the prescribed dynamics, ocean circulation and mixing will not respond to different melt/growth fluxes across the sensitivity simulations. Figure 8 shows how ice mass budget terms vary with snow volume across the simulations. The two hemispheres show fundamentally different behavior. With increasing snow volume, the total ice growth and melt are relatively insensitive in the Northern Hemisphere but increase considerably in the Southern Hemisphere. In the Southern Hemisphere, this is consistent with an increase in the ice volume annual cycle (Fig. 6b). However, in the Northern Hemisphere, the relative insensitivity of the total ice growth or ice melt to changing snow conditions appears to contradict the reduction in the ice volume annual cycle with increasing snow volume (Fig. 4b). This occurs because, on the hemispheric scale considered here, ice melt and growth happen throughout the year, and so changes in the magnitude of these terms are not necessarily associated with seasonal ice loss and gain.

In both hemispheres, the congelation ice growth declines with greater snow volumes (Fig. 8a and c), consistent with a greater insulating effect of the snow. This occurs in spite of lower air temperatures in the simulations with more snow (Fig. 7a and d), which are themselves partly a consequence of reduced heat conduction and hence lower surface heat loss

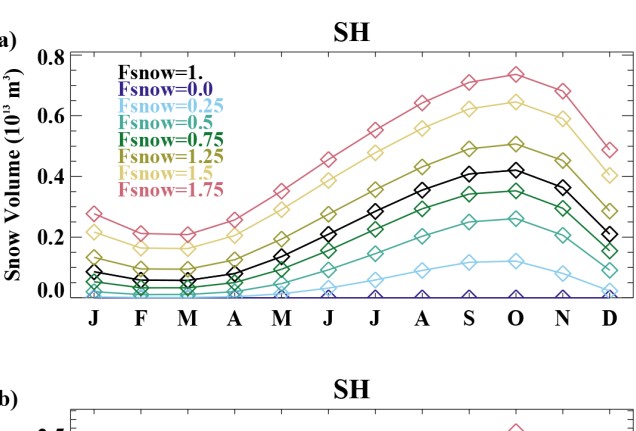

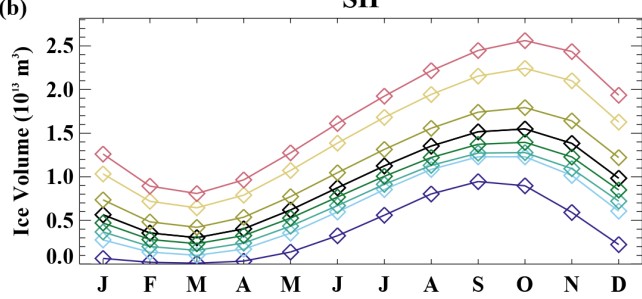

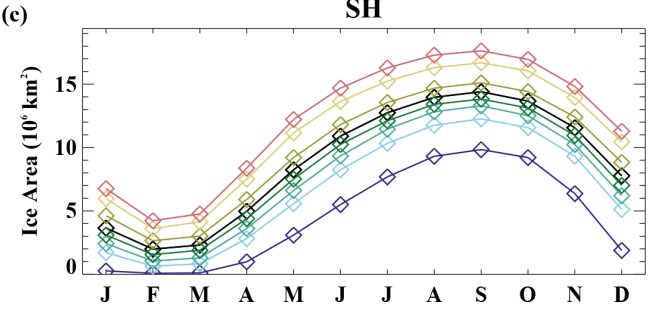

**Figure 5.** The Southern Hemisphere **(a)** snow volume, **(b)** ice volume, and **(c)** ice area in the different $F_{snow}$ experiments.

to the atmosphere. With thicker snow, the snow–ice formation increases in both hemispheres. Frazil ice formation remains quite similar across the runs. The dominance of these different ice growth changes differs considerably in the two hemispheres. In the Northern Hemisphere, changes in the ice mass budgets are quite small, and there is a rough balance between the decrease in congelation ice growth and increase in snow–ice formation. This leads to a weak sensitivity in total ice growth. However, in the Southern Hemisphere, the snow–ice formation changes are very large. They overwhelm the decreasing congelation growth, leading to a net increase in ice growth with increasing snow volume.

The simulations performed here reach an equilibrated state, and thus the changes in ice growth are balanced by changes in ice melt. In the Northern Hemisphere (Fig. 8b), there are reductions in the total surface ice melt with increasing snow volume. This is related to a later transition to a snow-free surface in the summer, a shorter snow free season,

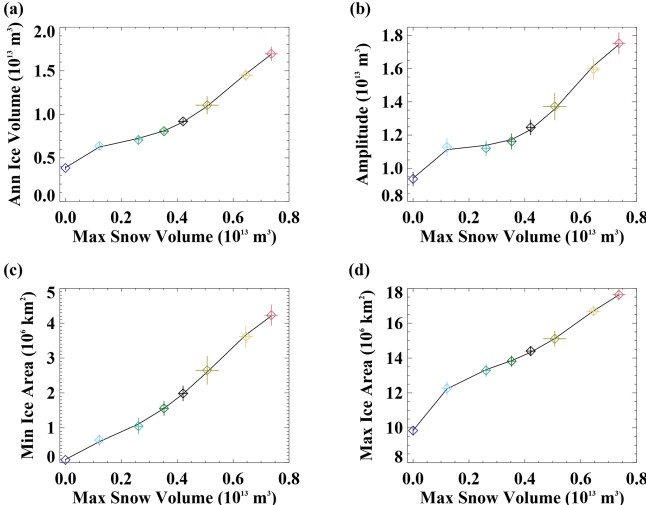

**Figure 6.** The relationship of the climatological Southern Hemisphere **(a)** annual ice volume, **(b)** amplitude of the ice volume annual cycle, **(c)** annual minimum ice area, and **(d)** annual maximum ice area versus the annual maximum snow volume. The diamonds show the simulated mean values, and whiskers are ± 1 standard deviation across the 20 years analyzed for each simulation. The colors the same as in other figures. The black line is a fourth-order polynomial fit.

and higher surface albedo. The total hemispheric basal ice melt generally increases with increased snow volume, which is largely a consequence of the greater ice area and larger seasonal ice cover. Notably, for the simulations with maximum hemispheric snow volume greater than $0.5 \times 10^{13}\,\mathrm{m}^3$ (the $F_{\mathrm{snow}} = 1.5$ and $F_{\mathrm{snow}} = 1.75$ cases), a large fraction (about 50 %) of the summer Arctic sea ice remains snow covered, with that ice experiencing no surface melt. The perennial snow cover in these simulations results in a consequently larger sensitivity of the total hemispheric ice volume to snow volume as seen in Fig. 4a.

In the Southern Hemisphere (Fig. 8d), lateral and surface melt are low across all simulations. Surprisingly, even with no snow on sea ice, the surface melt is only 7 % of the total annual melt. In the model simulations around Antarctica, much of the ice forms in coastal regions, and is transported away from the continent, where it subsequently melts from its base due to large ice–ocean heat exchange. This allows basal ice melt to occur year-round (Fig. 2b), and this term dominates even when no snow cover is present to protect the ice surface from melting. With increasing snow depth, the basal ice melt increases. This is not a direct response to the changing snow conditions but instead results from the mostly seasonal Antarctic ice and the larger volume of ice available to melt in the thicker snow cover simulations.

In order to better understand the differences in the sensitivity to snow cover across the Northern Hemisphere and Southern Hemisphere, it is illustrative to consider the average mass budgets in the perennial and seasonal ice pack re-

gions separately (Figs. 9 and 10). Here we define the perennial ice pack region as the area that has greater than 0.15 ice concentration at the annual ice area minimum, which occurs in September in the Northern Hemisphere and February in the Southern Hemisphere. The seasonal ice pack region encompasses the area where ice is present at greater than 0.15 ice concentration at the time of the annual ice maximum (March for the Northern Hemisphere and September for the Southern Hemisphere) but excluding the perennial ice region. The ice growth and melt budgets do not balance within these regional domains because ice can be transported between them. For the perennial ice region (Fig. 9a), congelation growth declines considerably and nonlinearly with increasing snow depths. This dominates the net growth sensitivity for Northern Hemisphere ice in the perennial zone and is balanced by ice melt that also decreases with increasing snow depths (Fig. 9b). The reduced ice melt at both the surface and base of the ice is consistent with the higher surface albedo, which reduces shortwave absorption within both the sea ice and ocean.

In the Southern Hemisphere (Fig. 9c), congelation ice growth also dominates the growth sensitivity when snow depths are less than 0.2 m. For deeper snow, increased snow–ice formation occurs, counteracting the decreasing congelation ice growth and leading to a limited total growth sensitivity. Southern Hemisphere ice melt within the perennial ice zone (Fig. 9d) shows very little sensitivity to snow. As indicated by the imbalance between melt and growth, most Southern Hemisphere ice within the perennial zone is lost through transport to the seasonal ice zone. In the Arctic, perennial ice is also lost through transport to the seasonal ice zone, but this plays a smaller role.

The net Northern Hemisphere seasonal ice zone growth (Fig. 10a) is smaller than that of the perennial ice zone in all experiments because the seasonal ice occurs in a region of warmer ocean and atmospheric conditions. The seasonal ice zone growth declines with increasing snow because of reduced congelation ice growth. The change in congelation growth per change in snow depth computed from the end-member simulations ($F_{\mathrm{snow}} = 0$ versus $F_{\mathrm{snow}} = 1.75$) is considerably larger for the seasonal ice (2.8 cm growth per centimeter of snow depth) than for the perennial ice (1.5 cm growth per centimeter of snow depth). This is consistent with thinner ice being more sensitive to the insulating effects of snow (Maykut, 1978). For the seasonal ice zone, the changes in congelation ice growth are largely compensated for by increased snow–ice formation for the thicker snow experiments, resulting in a limited sensitivity of the net growth to snow depth in those cases (Fig. 10a). The Southern Hemisphere seasonal ice zone net growth (Fig. 10c) is also smaller than that in the perennial ice zone (Fig. 9c), again due to warmer ocean and air in the seasonal ice region. The seasonal ice zone has declining congelation growth and increasing snow–ice formation with thicker snow depths (Fig. 10c). For simulations with thick snow cover, the snow–ice forma-

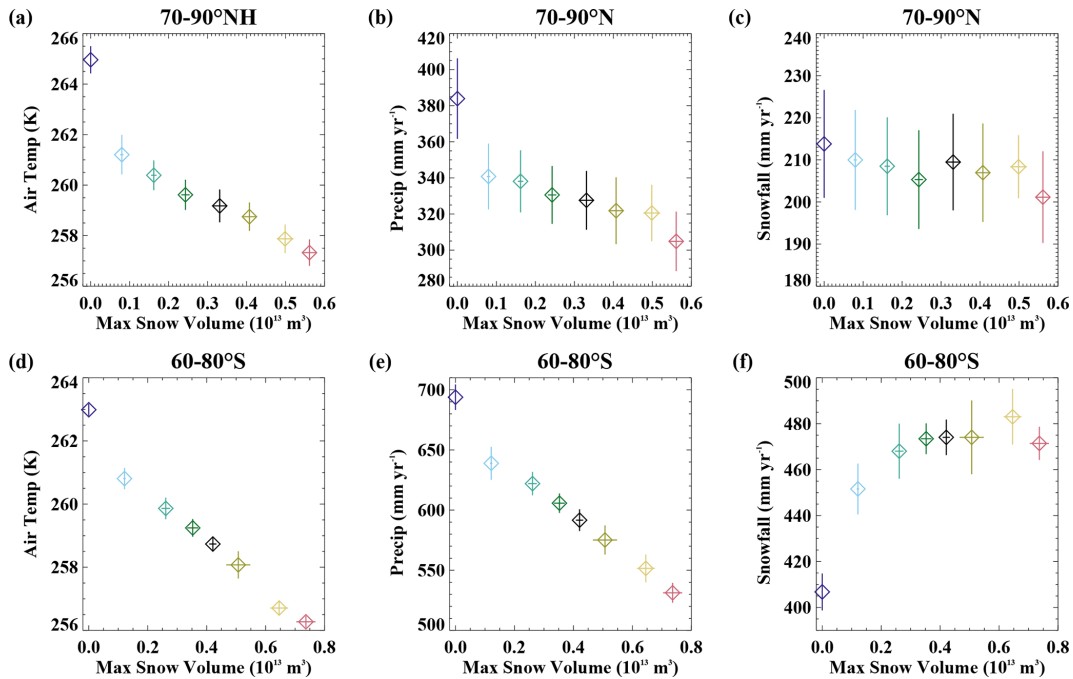

**Figure 7.** The relationship of the climatological 70–90° N **(a)** annual average surface air temperature, **(b)** annual total precipitation, and **(c)** annual total snowfall versus the annual maximum Northern Hemisphere volume of snow on sea ice across the simulations. Bottom panels show conditions for 60–80° S versus the Southern Hemisphere snow volume for **(d)** annual mean surface air temperature, **(e)** annual total precipitation, and **(f)** annual total snowfall. The whiskers show ± 1 standard deviation across the 20 years analyzed for each simulation.

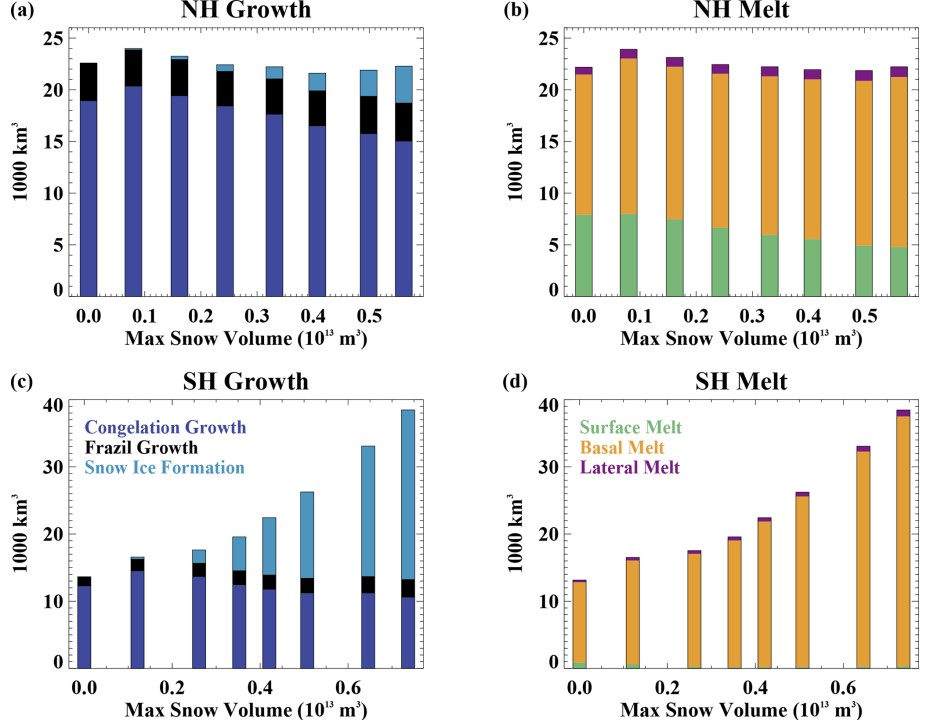

**Figure 8.** The volume of total annual **(a)** Northern Hemisphere ice growth, **(b)** Northern Hemisphere ice melt, **(c)** Southern Hemisphere ice growth, and **(d)** Southern Hemisphere ice melt as a function of the maximum snow volume across the simulations.

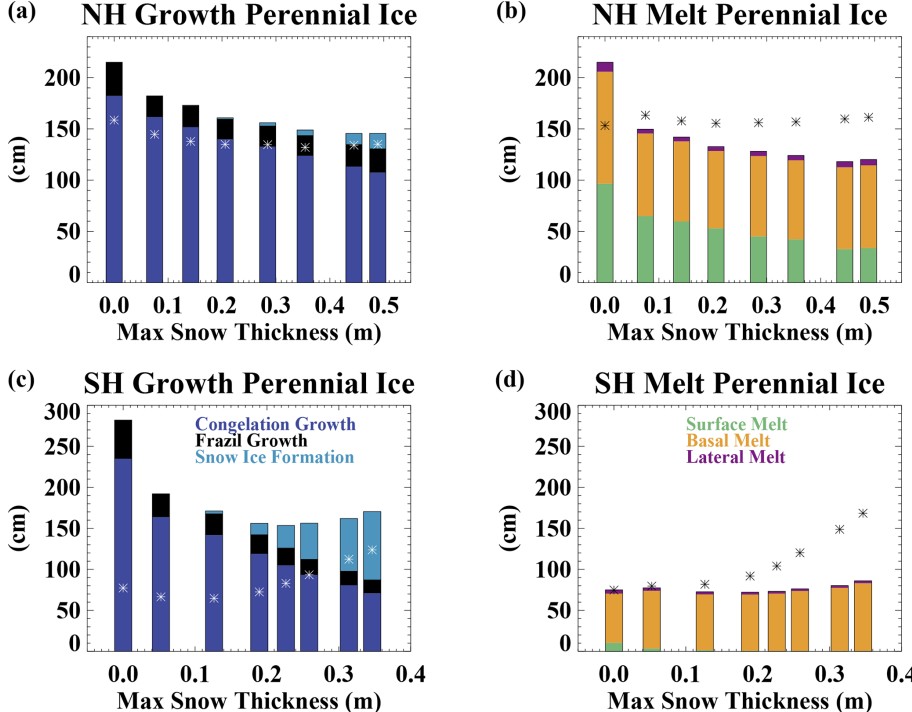

**Figure 9.** The perennial ice zone average **(a)** Northern Hemisphere ice growth, **(b)** Northern Hemisphere ice melt, **(c)** Southern Hemisphere ice growth, and **(d)** Southern Hemisphere ice melt as a function of the maximum perennial ice zone snow depth across the simulations. The asterisks show the net ice growth and melt for the seasonal sea ice domain for comparison (shown in more detail in Fig. 10). Note that the imbalance between the net ice growth and melt in an individual experiment results from ice motion that transports ice between the perennial and seasonal sea ice zones.

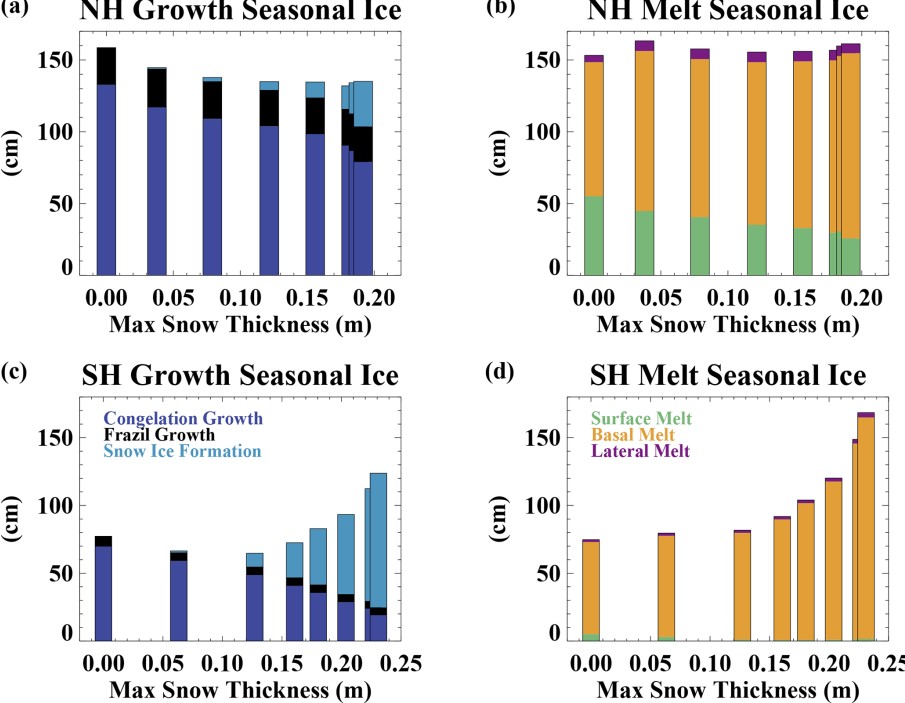

**Figure 10.** As in Fig. 9 but for the seasonal sea ice region.

tion dominates and leads to overall increases in ice growth. By definition, all seasonal ice is melted away every year. In both hemispheres, basal melting dominates, which is consistent with the influence of ocean heat availability in the sea-sonal ice zone.

The differences in the total hemispheric ice growth budgets in the two hemispheres (Fig. 8) can be explained by differences in the relative importance of perennial and seasonal ice and the changing mass budgets within those regions. In the Northern Hemisphere, ice growth declines with increasing snow depths in both the seasonal and perennial ice packs as a consequence of the insulating effect of the snow (Figs. 9 and 10). However, the amount of seasonal ice also declines (Fig. 11), and because that seasonal ice has reduced growth relative to the perennial ice, this counteracts the local insulating impact of the snow within each ice zone and causes a limited sensitivity of ice growth to snow depth for the hemispheric totals (Fig. 8). In the Southern Hemisphere, the dominance of seasonal ice in all experiments (Fig. 11) and increasing snow–ice formation with snow depth (especially in the seasonal ice zone) results in an increase in growth with increasing snow depth on the hemispheric scale.

### 3.4    Response in a 2xCO2 Climate

The hemispheric differences in the sea ice response to changing snow cover are suggestive of a climate state dependence. In order to explicitly test how the influence of snow on sea ice may change in a changing climate, we have performed an additional set of simulations in a climate state with atmospheric $CO_2$ concentrations that are twice the preindustrial value (2xCO2 runs). We use the same experimental design where the snowfall on sea ice is multiplied by a constant factor ($F_{snow}$; see Table 1). In the warmer 2xCO2 climate, more precipitation falls as rain, the rain-season duration increases, and consequently snowfall is reduced. Additionally, the area on which to accumulate snow is lower. These factors result in a smaller range of snow volume amounts across our sensitivity simulations. The Arctic is seasonally ice-free in all of the simulations performed, and the Antarctic has considerable reductions in ice area compared to the preindustrial runs.

The relationship between Northern Hemisphere sea ice properties and the maximum snow depth is shown in Fig. 12. With seasonal ice, higher air temperatures, and reduced snowfall present in the simulations, the snow accumulation is much reduced in the 2xCO2 climate, leading to a smaller variation in maximum snow volume across the simulations than in the preindustrial climate runs. Except for the no-snow case, the Northern Hemisphere ice volume, amplitude of the ice volume annual cycle, and minimum and maximum ice area are quite insensitive to the snow volume amount. The no-snow case does have significantly thinner ice, a lower amplitude annual cycle, and reduced maximum ice area. In the Southern Hemisphere (Fig. 13), the ice volume, ice volume

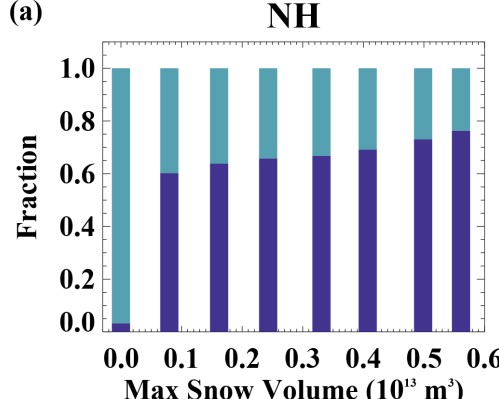

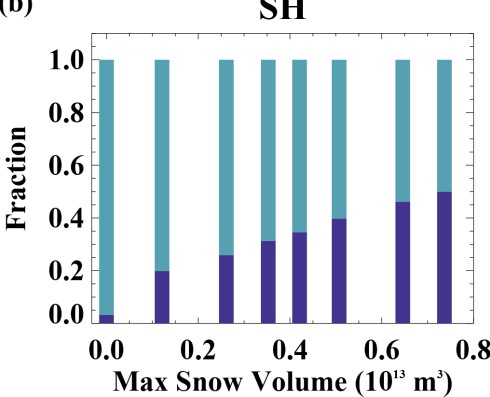

**Figure 11.** The fraction of the total ice area occupied by seasonal ice (light blue) and perennial ice (dark blue) as a function of the maximum snow volume across the experiments. Values are shown for the **(a)** Northern Hemisphere and **(b)** Southern Hemisphere.

seasonal amplitude, and ice area generally increase with increasing snow depths. This is quite similar to the results in preindustrial control simulations.

The hemispheric mass budgets associated with the changing ice conditions in the simulated 2xCO2 climate are shown in Fig. 14. In the Northern Hemisphere, congelation ice growth is smallest in the no-snow case and shows limited sensitivity to snow depth in the remaining simulations. The small growth sensitivity results from two largely compensating factors. With thicker snow, the insulating effect leads to reduced growth in the months when growth actually occurs. However, simulations with greater snow volume also have a lengthened ice growth season. This is a consequence of the albedo impact of the snow, which in the higher snow simulations contributes to later ice melt-out, less summer ocean shortwave absorption, and an earlier fall freeze-up. These sensitivities are indicative of the Arctic in a completely seasonal ice regime. The Southern Hemisphere mass budgets in the 2xCO2 climate (Fig. 14c and d) have a quite similar sensitivity to that of the preindustrial simulations, likely because of the prevalence of seasonal ice and snow–ice formation in both climate states. With increasing snow volumes, conge-

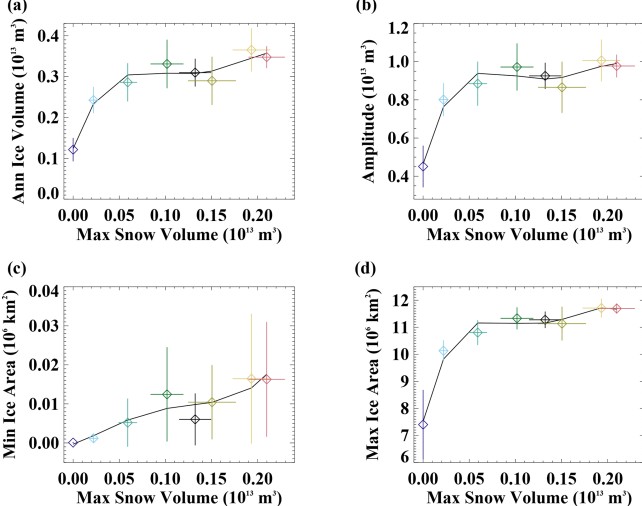

**Figure 12.** The relationship in the 2xCO2 simulations of the climatological Northern Hemisphere **(a)** annual ice volume, **(b)** amplitude of the ice volume annual cycle, **(c)** annual minimum ice area, and **(d)** annual maximum ice area versus the annual maximum snow volume. The diamonds show the simulated mean values, and whiskers are ± 1 standard deviation with the colors the same as in other figures. The black line is a fourth-order polynomial fit.

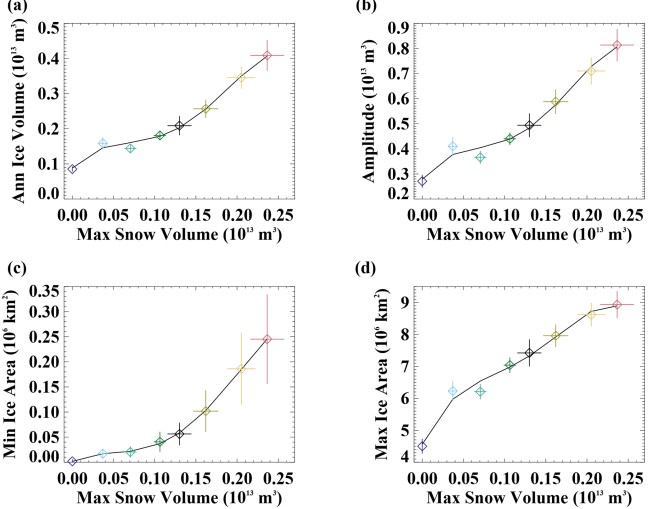

**Figure 13.** As in Fig. 12 but for Southern Hemisphere values in the 2xCO2 climate.

lation ice growth declines but is more than compensated for by increases in snow–ice formation, leading to increased total ice growth. This is balanced primarily by increased basal melt.

## 4   Conclusions

We have performed CESM2 coupled climate model experiments in preindustrial and 2xCO2 climate states with mod-

ified snowfall over sea ice. This has allowed us to isolate the influence of varying amounts of snow on sea ice in both hemispheres. In particular, we assess the influence on sea ice conditions and mass budgets. The model configuration used here includes a slab ocean model and prescribes the influence of ocean dynamics. Because of this, ocean dynamical feedbacks are not included although the ocean temperature can respond to changing surface fluxes. While previous studies have assessed the role of snow on sea ice, these have mostly been done in simpler frameworks including in the absence of coupled atmospheric feedbacks (e.g., Maykut and Untersteiner, 1971; Fichefet and Morales-Marqueda, 1999; Powell et al., 2005), for a perennial sea ice case (e.g., Maykut and Untersteiner, 1971), or for limited cases with simple models (Ledley, 1991). Our results do generally agree with these previous studies that increases in snow on sea ice result in a thicker ice cover and a cooler climate. Results from a coupled ice–atmosphere modeling study (Wu et al., 1999) show more complexity in the sea ice response to varying snow amounts and do not indicate a progressive increase in ice volume with increasing snow. We expect that this discrepancy with our results is due to model structural difference and, for example, the lack of a sub-grid-scale ice thickness distribution, simpler sea ice dynamics, and different dynamical ocean heat fluxes used in Wu et al. (1999). Nevertheless, as found here, Wu et al. (1999) also highlighted the importance of snow–ice formation for the Antarctic and that a disparate sensitivity is possible in the Arctic and Antarctic response to variable snow amounts.

For our sensitivity simulations in a preindustrial climate state in the Northern Hemisphere, the ice volume, ice area, and fraction of perennial ice generally increase with increasing snow depths. These changes are nonlinear with snow depth, consistent with previous studies (e.g., Maykut and Untersteiner, 1971). The Arctic atmosphere is colder in the simulations with increased snow on sea ice, reinforcing the sea ice changes. In the Arctic, our simulations can be roughly categorized into the three regimes of no snow, seasonal snow, and perennial snow cases. With no snow, the sea ice is thin, has a large annual cycle in ice volume, and is seasonally ice-free. In seasonal snow cases, with $F_{snow}$ from 0.25 to 1.25, the ice volume is relatively insensitive to snow depth, although the amplitude of the volume annual cycle does steadily decline. This suggests that for annual mean conditions, the insulating effect of the snow is roughly balanced by the albedo influence. In the more perennial snow cases ($F_{snow} = 1.5$ and 1.75), a considerable fraction of the ice remains snow covered year-round, and the ice is relatively thick. Increasing snow across the simulations results in a modest reduction in congelation ice growth and surface ice melt. However, the net hemispheric mass budgets are relatively insensitive to increasing snow. This is a consequence of the different mass budgets in seasonal and perennial ice zones and the changing perennial ice fraction across the simulations. In our simulations with increasing snow, there is a

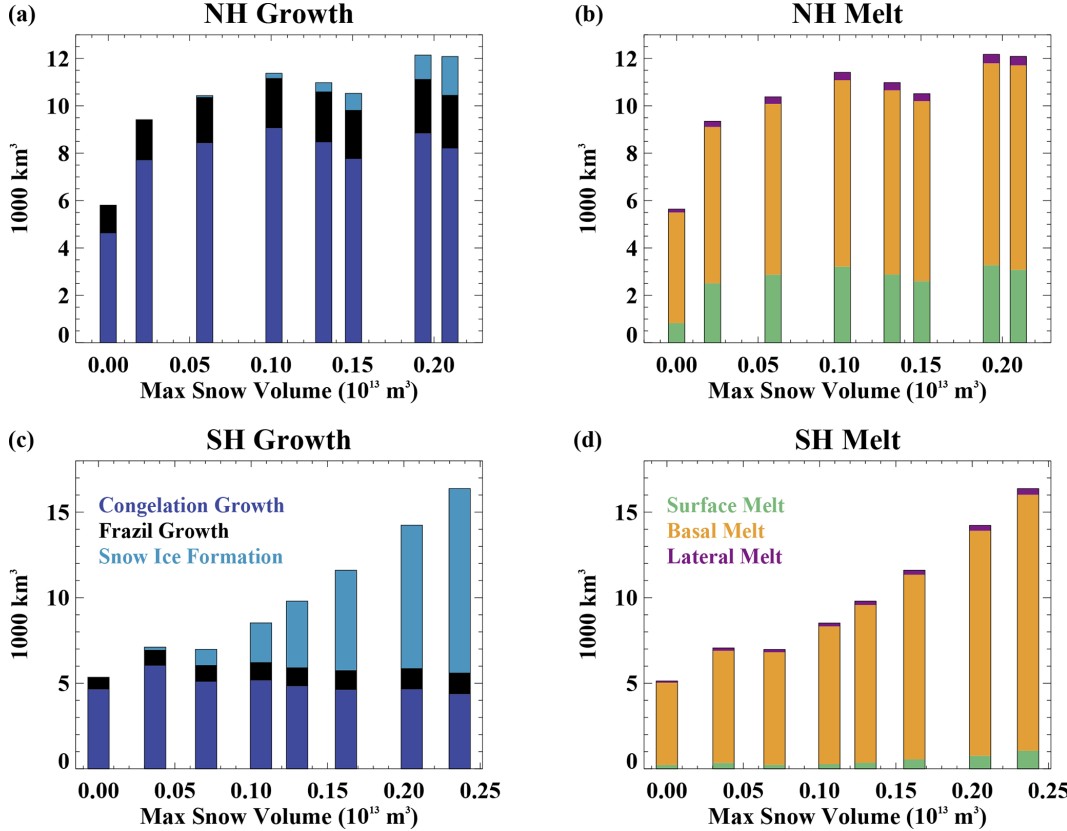

**Figure 14.** The volume in the 2xCO2 simulations of **(a)** Northern Hemisphere ice growth, **(b)** Northern Hemisphere ice melt, **(c)** Southern Hemisphere ice growth, and **(d)** Southern Hemisphere ice melt as a function of the maximum snow volume across the simulations.

significant reduction in growth of the perennial ice (consistent for example with Maykut and Untersteiner, 1971, and the insulating effect of the snow cover) but also a growing perennial ice fraction. Because perennial ice is primarily located within the Arctic basin, it has higher growth rates than seasonal ice that occurs in the more southerly marginal ice zone in which warmer ocean and atmosphere conditions are present. As such, the increase in perennial ice fraction counteracts the reduction of perennial ice growth, leading to a limited snow sensitivity for the hemispheric budgets.

In the Southern Hemisphere in a preindustrial climate, the ice volume and area also increase with increasing snow depths. In contrast to the Northern Hemisphere, the ice volume annual cycle amplitude increases. With increasing snow, the Antarctic experiences more ice growth and more ice melt. This occurs because of the prevalence of snow–ice formation in the Antarctic, which is a consequence of relatively thin ice and high precipitation. With increased snow, snow–ice formation increases and overwhelms decreases in congelation ice growth that result from the snow insulating effect. The high ocean heat content in the region allows this increased growth to be balanced by increased melting, which is almost entirely basal melt. The dominance of seasonal ice and high snow–ice formation results in a fundamentally different ice

mass budget sensitivity to snow cover in the Southern Hemisphere as compared to the Arctic.

Differences in the climate conditions between the poles are largely responsible for the differing sea ice response to changing snow amounts. This suggests that there may be a climate state dependence to our results. In order to assess this, we performed experiments with varying snow amounts in an equilibrated 2xCO2 climate. Except for the no-snow case, the Northern Hemisphere sea ice is quite insensitive to changing snow amounts. This may in part be due to the small range of snow accumulation that we are able to achieve with our experimental design because of the warm climate, low snowfall, and limited sea ice available as a platform to capture snowfall. All the 2xCO2 simulations have a seasonally ice-free Arctic, and all (except for the no-snow case) have similar congelation ice growth, which is counter to what would be expected due to the insulating effect of the snowpack. This is likely in part associated with a larger influence of ice thickness on heat conductivity in the case of thin snow cover. Nevertheless, monthly values of congelation ice growth do decrease with greater snow, but this is counteracted by a lengthened ice growth season leading to little sensitivity for the annual net growth. The longer ice growth season in simulations with more snow is in part a consequence

of the higher surface albedo that leads to a later ice melt-out, shortened ice-free season, and reduced accumulation of solar heat in the ocean over the summer months.

In the Antarctic, while the ice is thinner and less extensive in the 2xCO2 run, the sensitivity to changing snow amounts is qualitatively similar to those in the preindustrial climate state. In particular, with more snow, ice volume and area increase, and there is more ice growth as increases in snow–ice formation overwhelm decreases in congelation ice growth. We find little evidence that Arctic sensitivity to snow on sea ice will resemble the Antarctic in a warming climate. However, in both hemispheres, although the reasons differ, we find that more snow is a "friend" to sea ice rather than a "foe" (see Sturm and Massom, 2017) and results in increased ice volume and area and an overall cooler climate.

This work, along with previous studies, has highlighted the important role of snow on sea ice for the climate system. It has also indicated that this role can be quite complex with different factors dominating in different regions and under different climate regimes. This emphasizes the need to realistically simulate snow characteristics. However, most climate models, including the one used here, neglect some important aspects of snow on sea ice. For example, in the real system, snow is spatially heterogenous and wind can redistribute snow across different ice types or into leads (e.g., Sturm et al., 2002b; Leonard and Maksym, 2011). This influences the snow mass budget and the effective thermal and optical properties of the ice cover. Additionally, many climate models use constant values for properties such as snow thermal conductivity and albedo (or the factors from which albedo are computed) although field data indicate that these properties vary (e.g., Sturm et al., 1997, 2002a; Colonne et al., 2011). Parameterizations (e.g., Lecomte et al., 2013, 2015) and more sophisticated snow models (e.g., Liston et al., 2020) have been developed that account for some of these snow complexities. However, these are often not incorporated into global climate models. Additionally, ongoing work is needed to better understand the processes driving snow redistribution and metamorphosis, among other factors. Analysis of field data, such as from the year-long MOSAiC (Multidisciplinary drifting Observatory for the Study of Arctic Climate; Nicolaus et al., 2021) drift campaign, and remotely sensed information from campaigns like Operation IceBridge (e.g., Koenig et al., 2010) and ICESat-2 and CryoSat-2 (e.g., Kwok et al., 2020) will help to better constrain factors controlling snow distributions and properties and can enable improved representation of those processes into climate models. This will ultimately improve predictions of these important aspects of the changing polar regions.

*Code and data availability.* The CESM2 model used for the experiments described here is publicly available for download from https://www.cesm.ucar.edu/models/cesm2/release_download.html (last access: 5 June 2020; Danabasoglu et al., 2019). Post-processed data from the experiments described here for the climatological sea ice state, mass budgets, and polar atmosphere conditions are available from https://doi.org/10.5281/zenodo.5572930 (last access: 15 September 2021).

*Supplement.* The supplement related to this article is available online at: https://doi.org/10.5194/tc-15-1-2021-supplement.

*Author contributions.* MMH, BL, and MS devised the experiments. MMH performed the experiments and analyzed the simulations. All authors contributed expertise on snow and sea ice processes. All authors contributed to the writing of the manuscript.

*Competing interests.* The contact author has declared that neither they nor their co-authors have any competing interests.

*Acknowledgements.* We acknowledge computing resources (https://doi.org/10.5065/D6RX99HX) provided by the Climate Simulation Laboratory at NCAR's Computational and Information Systems Laboratory, sponsored by the National Science Foundation and other agencies. We thank three anonymous reviewers who provided very useful comments on the manuscript. This material is based upon work supported by the National Center for Atmospheric Research, which is a major facility sponsored by the National Science Foundation (NSF) under cooperative agreement no. 1852977. CE1

*Financial support.* Marika M. Holland and Laura Landrum received support from NSF grant no. OPP-1724748; Donald Perovich, Chris Polashenski, and David Clemens-Sewall received support from NSF grant no. OPP-1724540; Bonnie Light and Madison Smith received support from NSF grant no. OPP-1724467; Melinda Webster received support from NASA grant no. 80NSSC21K0264. CE2

*Review statement.* This paper was edited by John Yackel and reviewed by three anonymous referees.

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

## Remarks from the language copy-editor

CE1 Please confirm changes.
CE2 Please confirm changes.

## Remarks from the typesetter

TS1 Please give an explanation of why this needs to be changed. We have to ask the handling editor for approval. Thanks.
TS2 Please give an explanation of why this needs to be changed. We have to ask the handling editor for approval. Thanks.