# Peer review of "The influence of snow on sea ice as assessed from simulations of CESM2"

_The Cryosphere, 2021_

## Referee Comment (RC1)

**Review for "The influence of snow on sea ice as assessed from simulations of CESM2" written by Marika M. Holland et al..**

Snow plays a critial role in the Earth's climate system due to its high albedo and effective thermal insulating effect. Its presence can delay sea ice growth in winter and surface melting in summer, and also can contribute significantly to the sea ice mass through formation of snow-ice which is a major contributor for sea ice mass in the Antarctic and becomes more and more important in the Arctic. It is also a difficult parameter in the model to be represented because of lack of obsevations in the poles. The article written by Holland et al. accessed the influence of snow on sea ice in experiments using the Community Earth System Model, version 2 for a pre-industrial and a 2xCO2 climate state, and found that the sea ice in the Arctic and Antarctic responses differently to increasing snow accumulation on sea ice.  Through their study, Holland et al. underscore the importance of accurately representing snow accumulation on sea ice in coupled earth system models. The article is well structured, and well written. Snow is a hot topic in the recent studies and the articl is worth to be published. Thus I recommend publish this article after some minor revison.

General comments:
For figures, I suggest to set same limit for the y-axis for easy comparison, for example, Figure 2, using  [-220, 220] for y-axis whether in the Antarctic and Arctic. Similar for the other figures.

Specific Comments:
Line 15-16: To check "increasing snow results in a decrease in both sea ice growth and sea ice melt". From Figure 8, the decrease in congelation ice growth and basal ice melting is correct, but it is not ture for the other type of ice growth and melting.

Line 39: Snow ice was observed in Arctic (e.g., Granskog et al., 2017, JGR) and potentially becomes more and more common in the Arctic (Merkouriadi et al., 2020, GRL). Please add one statement about snow ice in the Arctic also.

Line 70-71, Rewrtite this sentence to make it more understandable.

Line 76-77: Wind-driven blowing snow into leads leads to snow loss on sea ice in the Antarctic, whici is typically not represented in climate models. This means that snow in the climate models tends to be overestimated, right? Could you disccuss a little bit about this effect affect the snow simulation in the CESM2 and how it affects your results in the whole article?

Line 148-149: citation of references with earlier publish year in front

Line 180, Change "make small contributions" to "only contribute slightly"

Line 195-197: It is a too long sentence, hard to udnerstand. Suggest to rewrite it.

Line 226, remove "( "and ")", and for before Fsnow. And how about for Fsnow 1.5 and 1.75

Line 229: Please check it is Fig. 3a or Fig. 3b?

Line 231: Refer to Fig. 3C aslo? It is much easier to see in Fig. 3c.

Line 257: "This will feedback" is difficult to understand for me. Could you rewrite it?

Line 259-260: citation references with earlier publish year in front

Line 302: remove "that is present"

Line 323-324, Combine "This dominates …. This is balanced by" two sentences into one.

Line 410: For Figure 11, set same upper limit for x-axis for the SH and NH

---

## Referee Comment (RC2)

**Review: tc-2021-174, *"The influence of snow on sea ice as assessed from simulations of CESM2"*, by Marika Holland et al.**

**1. GENERAL COMMENTS**

This work is of considerable importance. It addresses a major gap in our understanding of Earth's polar sea-ice environments and their likely trajectory in a warming climate – namely (change and variability in) the role of snow accumulation. At the same time, it provides new insight into the climatic importance of snow on sea ice, and identifies and highlights a number of important feedbacks in the system. Another important factor is that the paper directly compares results from both the Arctic and Antarctic, and highlights different hemispheric responses of sea ice in the two hemispheres as a result of differing simulations in snow accumulation in both Pre-Industrial and $2xCO_2$ climate scenarios. This underlines the critical importance of accurately representing snow accumulation in Earth system models, and lays the groundwork for important future work towards more accurate representation of the cryosphere and cryospheric processes in such models.

The article is generally well written, and lays out its scientific rationale, discussion and conclusions in a clear, concise and well-structured fashion. There are relatively few grammatical and punctuation errors, but a number of ambiguous statements and uses of modeling jargon that may challenge the general reader (see SPECIFIC COMMENTS below) - but these can be easily fixed. Re the figures, these are largely informative and well presented, but I'm afraid that I'm colour blind so struggled with some of them – notably Figures 4, 6 and 7. The paper is also well referenced, with an appropriate number and quality of references.

I have a number of questions/issues regarding the methods used and the results and their interpretation – that I feel need addressing before the paper can be published. First up, it's not clear how important snow properties (apart from thickness) – such as density, thermal conductivity and albedo - are treated/parameterised in the model, if at all. Other issues relate to the treatment of wind-blown snow removal/redistribution, and the conversion of snow to snow ice – the Methods section needs more information on the how snow is treated/parameterised in the model. Also, current observations of the thickness distributions of Antarctic sea ice and its snow cover are inadequate to validate model results such as those shown here e.g., in Figure 1d-e. A further question relates to the use of 70-90 deg as the geographical domain for both the northern and southern hemispheres i.e., in Figure 7. I can understand this for the Arctic, but not for the Antarctic – where 70-90 deg S mainly covers the Antarctic ice sheet and only partially covers the Antarctic sea-ice zone. There's also an apparent discrepancy between the contributions of frazil ice to the mass balance in the model output (low) versus that which is typically observed (high). These and other questions/issues are outlined in SPECIFIC COMMENTS below.

Moreover, the Conclusions section would benefit from discussion of possible caveats and limitations in this study, and also future work that could or needs to be carried out in both observation and modeling. This could again highlight key gaps in our understanding of snow in the sea ice systems, the seasonal, regional and hemispheric dependence of the relationships, and the need for more large-scale observations of snow thickness and properties as well as accurate precipitation rates (see Webster et al., 2018). It should also address the snowfall v accumulation discrepancy factor (due to wind-blown snow redistribution/loss).

Given the importance of this work, the substantial large-scale conclusions reached and the fact that it represents substantial progress beyond current scientific understanding, may I recommend publication pending the authors addressing both the minor and more major comments/issues. In all cases, I hope that my comments/suggestions help.

**2. SPECIFIC COMMENTS**

2.1 The Introduction puts the study and its importance nicely into context, including previous work in the field. Possibly one thing lacking is equal coverage in the Introduction of snow on Arctic and Antarctic sea ice, and important properties and attributes therein from the perspective of this study. Notably, the 3 paragraphs from lines 40 to 69 almost exclusively focus on the Arctic. Please provide more information on what is known – and what is not known – about Antarctic snow on sea ice, and how it is relevant to this study.

2.2 Line 21 – what are these competing processes and feedbacks?

2.3 Line 34 – the assumption here is that sea ice thickens by thermodynamic growth alone, whereas dynamics are also important – this needs stating/clarifying here.

2.4 Line 36 – high snow albedo also reduces solar heating of the underlying ocean.

2.5 Line 48 – needs Andreas and Ackley reference.

2.6 I didn't fully understand Lines 53-57.

2.7 Regarding treatment of snow on Antarctic sea ice and the effects of snow on sea ice simulations - may I also suggest referring to: Wu et al. 1999 (Wu, X., W.F. Budd, V.I. Lytle, and R.A. Massom. 1999. The effect of snow on Antarctic sea ice simulations in a coupled atmosphere-sea ice model. *Climate Dynamics*, 15, 127-143).

2.8 Line 80 - State that SHEBA is an Arctic campaign.

2.9 Lines 87-88 – as stated, another important part of this question is how snow affects the retreat and duration of sea ice coverage, and whether there is regional and hemispheric dependence.

2.10 While the emphasis and focus is on snow accumulation/thickness, it is not clear from the Methods Section how snow properties (apart from thickness) are treated/parameterised in the model – if at all. These properties include snow density, related thermal conductivity, and grain size as it affects albedo/light transmission etc. What values are used for the different snow parameters (including albedo)? Also – how is snow converted into snow ice? Please include more information in the Methods section on how the snow is treated, and the possible caveats/limitations.

2.11 By the same token, it is not clear whether Antarctic snow is parameterised differently to Arctic snow in the model. What are the differences between the physical, optical and thermal properties of snow on Arctic versus Antarctic sea ice, and do they make a difference to (the modelling of) sea ice mass balance and areal coverage? – and if so, how?

2.12 Also, it's not clear how the model treats or accounts for any discrepancy between snowfall and snow accumulation, given horizontal redistribution of the snow by winds. Sorry if I'm missing something here, but I'm just wondering whether wind speed is taken into account in the model re the different climate state scenarios, and whether increasing (change in) wind speed can also be tested as an additional factor affecting snow accumulation. This point is stimulated by the tendency of snowfall over Antarctic sea ice to typically occur under stormy, windy conditions – with wind-blown horizontal redistribution being a major process determining the snow thickness distribution and also snow loss into leads. In other words, snowfall does not equal snow accumulation. This factor is acknowledged in lines 75-77 of the Introduction – but it's again not clear how or whether this "discrepancy factor" is accounted for here and, if not, whether this is an issue.

2.13 Also - while snowfall may increase in a warmer climate scenario, will this be compensated by increased wind-blown "loss" (including sublimation) in terms of snow accumulation on sea ice?

2.14 These additional snow factors/properties – on top of snow accumulation alone – may potentially be important in terms of their effects on sea-ice mass balance.  My suggestion would be to state this in the Introduction (as possible caveats), then revisit in the Conclusions i.e., state there that the results presented here are based on accumulation only, and that more work is required (if this is the case).  This would make a more convincing case for focussing on snow accumulation and thickness alone here.  Maybe future work would/could involve sensitivity studies to account for what is currently known about Arctic and Antarctic snow physical and optical properties on sea ice.

2.15 Regarding these comments, may I suggest that the Methods section focus more on snow, how it is modelled here, and why that approach is taken.  By the same token, the paper would benefit from providing more details on the nuts and bolts of the sea ice, atmosphere and ocean components of the model, and how hemispheric differences are catered for – in a Supplementary Section.  While relatively concise, the current description provided in Lines 107-114 could be expanded upon in a Supplementary Section, to also remove jargon and aid/enhance the reader's understanding.  It could also highlight current strengths and weaknesses of the model; this information is currently lacking.

2.16 Lines 108-114 – not clear to a non-modelling person – jargon.

2.17 Line 121 – what environmental conditions?  Needs more detail.

2.18 Lines 126-128 – I read this a number of times but still didn't fully understand.

2.19 Line 130 – what atmospheric feedbacks?  Give details here.

2.20 Lines 144-147 – these statements need backing up with references.

2.21 Lines 148-149 – it could be argued that current knowledge of large-scale sea ice thickness and its variability and change in space and time is very poor indeed and inadequate (see IPCC SROCC report etc.).

2.22 Line 154-155 – needs backing up with a reference.

2.23 Lines 158-161, Figure caption – make it clear here whether these are observed or modelled.

2.24 In the Results, one thing that struck me about Fig 2b – the SH mass budget control run – is the relatively low contribution of frazil ice (compared to congelation ice), as discussed in Lines 182-192.  This is different to what is typically observed around Antarctica, with a relatively high proportion of frazil due to the highly-dynamic and turbulent conditions there.  See for example: http://aspect.antarctica.gov.au/home/about-sea-ice/ice-formation (based on Worby et al., 1998):  "Analysis of 173 cores taken on six voyages into the East Antarctic pack between 1991 and 1995 revealed that on average the pack was comprised of 39% columnar ice, 47% frazil ice and 13% snow ice, with other ice types making up the remaining 1%. These figures indicate the importance of the dynamic processes within the pack, which favour the growth of frazil ice. Snow ice is also seen to make a significant contribution to the total ice mass of the region."  See also Lange and Eicken 1991 - (Lange, M., & Eicken, H. (1991). Textural characteristics of sea ice and the major mechanisms of ice growth in the Weddell Sea. *Annals of Glaciology, 15*, 210-215. doi:10.3189/1991AoG15-1-210-215). I'm not quite sure what to suggest here, given the discrepancy between this general understanding of the composition of Antarctic sea ice and the model results vis a vis the contribution of frazil ice.

2.25 Line 202 – I didn't understand this sentence.

2.26 Line 231 – what exactly are these different factors?  Also, this is rather ambiguous, as different factors control ice edge location in different regions – at least around Antarctica.

2.27 Line 234 – what is meant by "reduced annual cycle"?

2.28 Lines 294-296 – these findings are similar to those of Wu et al. 1999 (Wu, X., W.F. Budd, V.I. Lytle, and R.A. Massom. 1999. The effect of snow on Antarctic sea ice simulations in a coupled atmosphere-sea ice model. *Climate Dynamics*, 15, 127-143).

2.29 Line 306 – why is lateral melt low across all Antarctic simulations? Lateral melt is thought to be a major factor in the annual meltback of Antarctic sea ice (numerous papers by Gordon etc.).  Once again, it should probably be stated that ice formation/advance and retreat/meltback are driven by not only thermodynamics but also ice dynamics.

2.30 Lines 308-309 – how realistic is this assertion that most Antarctic ice forms in coastal regions?  Other studies have shown that formation within the pack and at the ice edge are also very important, depending on region.

2.31 Line 376 – does rainfall occur over the entire sea-ice zones, and is this rainfall area alos seasonally dependent?  Also, does rainfall remove an existing snow cover, or change its albedo?

2.32 Lines 415-421 – May I also suggest including discussion of the new results in the light of the Wu et al. (1999) paper in paragraph 1 of the Conclusions i.e., comparing findings of that study with this one.  This also relates to Lines 440-447.

2.33 Line 425 – doesn't this also depend on ice concentration?  i.e., a lower concentration or area of sea ice coverage enhancing air temperatures.

2.34 Lines 435-438 – I didn't quite follow this argument of higher growth rates under thicker perennial ice.  Does not congelation ice growth decrease with increasing ice thickness?  Also, what about rapid frazil ice formation in more turbulent conditions of the marginal ice zone?  Moreover, the conceptual model proposed is again only based on consideration of thermodynamics.  How does the model account for dynamic thickening by deformation?

2.35 Line 466 – suggest changing to: "….is a "friend" to sea ice rather than a "foe" (see Sturm and Massom, 2017)…." NB This reference is also incomplete in the Reference List, missing the chapter title.

2.36 Page 22 – May I suggest that the Conclusions section ends with discussion of possible caveats and limitations in this study, and also future work that could or needs to be carried out in both observation and modelling.  This could reiterate key gaps in our understanding of snow in the sea ice systems, the seasonal, regional and hemispheric dependence of the relationships, and the need for more large-scale observations of snow thickness and properties as well as accurate precipitation rates (see Webster et al., 2018).  It should also address the snowfall v accumulation discrepancy factor (due to wind-blown snow redistribution/loss).

2.37 Page 11 – probably better to add full figure captions to Figures 5 and 6.

2.38 General question - What impact does (changing) snow accumulation have on the timing of sea ice advance and retreat, and the resultant duration of annual coverage in both polar regions?

**3. TECHNICAL CORRECTIONS**

3.1 Consistency – use either snow depth or snow thickness throughout.

3.2 Line 19 – grammar

3.3 Line 24 – reference needed.

3.4 Lines 258-259 – grammatical error.

---

## Referee Comment (RC3)

Review of "The influence of snow on sea ice as assessed from simulations of CESM2" by Marika M. Holland et al

General comments:

The impact of snow on sea ice is difficult to assess due to the relative lack of long-term observations in remote polar regions. Given the many, competing feedbacks between snow on sea ice and sea ice itself, it is important to quantify these feedbacks and their climate impacts. In my view, this paper significantly adds to our current understanding of snow on sea ice and its impacts. It presents a study of the influence of snow thickness on sea ice in both pre-industrial and 2xCO2 experiments in coupled slab-ocean CESM2, varying the snow thickness by multiplying the snowfall on sea ice by different constant factors. The paper finds that overall, in both hemispheres, snow on sea ice tends to result in increased sea ice volume and cooler temperatures, although the ice mass budget response differs between the hemispheres. In a 2xCO2 climate, the study finds that Arctic sea ice sensitivity to snow depth is reduced, whereas the Antarctic sensitivity is similar to the pre-industrial climate.

The paper is overall well-structured, clear, and well-written, and presents results that, in my view, are of high scientific interest. It is well-referenced, with an appropriate number of references. There is a good number of figures, and the figures are generally clear, although I have a few minor suggestions below. There are some parts of the paper where I think some additional detail would be helpful, (as explained in my comments,) but overall, I strongly recommend that this paper be accepted for publication, with minor revisions.

Specific comments:

My first suggestion pertains to the use of the slab-ocean model (SOM); I recommend that the authors discuss some of the associated caveats in more detail in the paper. (To be clear, I do not think the use of a SOM is inappropriate for this study; I simply would encourage the authors to expand some of the discussion of the SOM.) The paper mentions that the SOM used has a prescribed ocean heat flux convergence from the CESM2 preindustrial control (line 113-114). However, it has been found that ocean heat convergence weakens near the ice edge with increased CO2 (as mentioned in Bitz et al 2005, which this paper cites) and thus, I wonder if the Qflux used in this configuration may still retain some bias related to this. Also, the lack of representation of ocean dynamics in the SOM is briefly mentioned, but I think it would be helpful to briefly discuss possible impacts the lack of ocean dynamics could have on the results of this study, where applicable (eg. possibly in terms of hemispheric differences being modified by ocean dynamics).

Secondly, although several snow processes are mentioned, it is not always clear which of these processes are represented in the model (eg. wind-driven blowing snow, rain-on-snow, snow density and thermal conductivity, etc.). I think it would be helpful to include a brief description of which snow processes and properties are described by the model, and possibly what biases could be present from the exclusion of certain processes. (It seems to me that this is briefly discussed in places, but I think more detail would be beneficial.)

Finally, I think there could be additional discussion included in the conclusion about what uncertainties remain, and possible next steps.

Specific comments line-by-line:
(N.b.: These comments may be taken as suggestions .)

105: Is this due to processes not being represented in the model?

116: Can you be more specific about what "mostly equilibrated" means in this context?

117: Perhaps describe how missing ocean dynamical feedbacks could impact the results here.

120: It would be helpful to include some description of the snow processes in the model in or before this paragraph (or somewhere else in the section, where appropriate).

145-146: There seems to be a missing citation here for "lack of significant anthropogenic ice loss that has been observed in recent decades"

Figure 1: If possible, I suggest that these map figures be made without the grid, since it is too faint to be visible and it seems to be producing artefacts. Also, for clarity, consider specifying the spacing of the black isolines somewhere (perhaps in the figure caption).

Also, although snow observations are limited, I think it could be beneficial to include a comparison with snow thickness observations if those are available; perhaps as a supplementary figure if it does not fit well within the main text.

164: Are there any ice processes not simulated by the model that could have an impact on the results?

172: Some points that may be helpful to discuss briefly: how might the constant density impact the results? In what regions would we expect results to be most impacted by this density assumption?

Figure 3,5: Consider aligning the months here to the seasons as in Fig. 2.

273: The term "basically equivalent" is vague; could you clarify what do you mean by this?

Figure 10: Some of the overlapping bars are difficult to see; is it possible to make them narrower so that they overlap less?

331 (and other places where ice motion is mentioned): It would be helpful to mention what drives the ice motion in this model.

426: This is the first time in this paper that these cases are specifically referred to as regimes so it seems to come somewhat out of nowhere; consider mentioning it earlier in the paper.

Technical corrections/notes:

111: I think "mixed-layer-averaged" should be hyphenated, or this could be rephrased to avoid hyphenation (the wording is somewhat ambiguous as-is)
117: Missing comma after "excluded"
173: The 3 should be a superscript in $kg/m^3$
212: I think "snow free" should be "snow-free"
224: It would be clearer if this said something like "a significant jump in ice volume and area from the Fsnow=0 case to the Fsnow=0.25 case"; clarifying that the jump is between the 0 and 0.25 cases.
240: Missing word, should be "the colors are the same"
258: Hyphen missing in "high-latitude"
259: Missing word, should be "consistent with a reduced"
300: As in 212, I think this should be "snow-free"
304: Hyphen missing in "snow-covered"
341: I think there should be a comma before "but"
349,363: Hyphen missing in "snow-ice" (assuming that the convention being used is to hyphenate)
379,415,422, etc.: Inconsistencies in hyphenation of "preindustrial" vs "pre-industrial"
References (general): Many entries here include italicization, but TC guidelines indicate to not italicize text in references.

---

## Author Comment (AC1)

Response to reviews.

Reviewer 1.

Snow plays a critial role in the Earth's climate system due to its high albedo and effective thermal insulating effect. Its presence can delay sea ice growth in winter and surface melting in summer, and also can contribute significantly to the sea ice mass through formation of snow-ice which is a major contributor for sea ice mass in the Antarctic and becomes more and more important in the Arctic. It is also a difficult parameter in the model to be represented because of lack of obsevations in the poles. The article written by Holland et al. accessed the influence of snow on sea ice in experiments using the Community Earth System Model, version 2 for a pre-industrial and a 2xCO2 climate state, and found that the sea ice in the Arctic and Antarctic responses differently to increasing snow accumulation on sea ice. Through their study, Holland et al. underscore the importance of accurately representing snow accumulation on sea ice in coupled earth system models. The article is well structured, and well written. Snow is a hot topic in the recent studies and the articl is worth to be published. Thus I recommend publish this article after some minor revison.

Thank you for your helpful comments on our manuscript.

General comments:
For figures, I suggest to set same limit for the y-axis for easy comparison, for example, Figure 2, using [-220, 220] for y-axis whether in the Antarctic and Arctic. Similar for the other figures.

Thank you for the suggestion. We have remade the figures where possible to use the same y-axis for equivalent variables in the Antarctic and Arctic to enable better comparison.

Specific Comments:
Line 15-16: To check "increasing snow results in a decrease in both sea ice growth and sea ice melt". From Figure 8, the decrease in congelation ice growth and basal ice melting is correct, but it is not ture for the other type of ice growth and melting.

We have clarified that this refers to a decrease in congelation growth and surface ice melt. We also now mention that there is a difference between areas of perennial ice (where these factors dominate) and areas of seasonal ice.

Line 39: Snow ice was observed in Arctic (e.g., Granskog et al., 2017, JGR) and potentially becomes more and more common in the Arctic (Merkouriadi et al., 2020, GRL). Please add one statement about snow ice in the Arctic also.
We have now added a statement regarding snow ice in the Arctic with these citations.

Line 70-71, Rewrtite this sentence to make it more understandable.
We have simplified this sentence by taking out the initial clause.

Line 76-77: Wind-driven blowing snow into leads leads to snow loss on sea ice in the Antarctic,

whici is typically not represented in climate models. This means that snow in the climate models tends to be overestimated, right? Could you disccuss a little bit about this effect affect the snow simulation in the CESM2 and how it affects your results in the whole article?

We have added a statement here that this means the models are "missing a potentially important Antarctic snow sink and may overestimate snow-ice formation.". We further elaborate about the impacts on the net sea ice mass budgets: "However, the net impact on sea ice mass budgets is unclear since much of the snow lost to leads may result in ocean supercooling and ice growth."

Line 148-149: citation of references with earlier publish year in front
Fixed

Line 180, Change "make small contributions" to "only contribute slightly"
Changed as suggested.

Line 195-197: It is a too long sentence, hard to udnerstand. Suggest to rewrite it.
We have split this into two sentences to make it easier to understand.

Line 226, remove "( "and ")", and for before Fsnow. And how about for Fsnow 1.5 and 1.75
We have removed the parentheses here as suggested.

Results for the Fsnow 1.5 and 1.75 cases are discussed in the last sentence of this paragraph. We have added some clarifying text to this sentence to emphasize that this is for the Fsnow 1.5 and 1.75 cases.

Line 229: Please check it is Fig. 3a or Fig. 3b?
Thank you for catching this. It should indeed be Fig. 3b and is now fixed.

Line 231: Refer to Fig. 3C aslo? It is much easier to see in Fig. 3c.
Thank you for this suggestion. We now also reference Figure 3c here.

Line 257: "This will feedback" is difficult to understand for me. Could you rewrite it?
This has been rewritten to clarify that the atmosphere changes will influence ice-atmosphere heat exchange.

Line 259-260: citation references with earlier publish year in front
Fixed

Line 302: remove "that is present"
Removed as suggested.

Line 323-324, Combine "This dominates …. This is balanced by" two sentences into one.
The sentences have been combined into one.

Line 410: For Figure 11, set same upper limit for x-axis for the SH and NH
We will set the same upper limit here and in similar figures.

---

## Author Comment (AC2)

Response to Reviewer 2.

**Review: tc-2021-174, "*The influence of snow on sea ice as assessed from simulations of CESM2*", by Marika Holland et al.**

**1. GENERAL COMMENTS**

This work is of considerable importance. It addresses a major gap in our understanding of Earth's polar sea-ice environments and their likely trajectory in a warming climate – namely (change and variability in) the role of snow accumulation. At the same time, it provides new insight into the climatic importance of snow on sea ice, and identifies and highlights a number of important feedbacks in the system. Another important factor is that the paper directly compares results from both the Arctic and Antarctic, and highlights different hemispheric responses of sea ice in the two hemispheres as a result of differing simulations in snow accumulation in both Pre-Industrial and 2xCO2 climate scenarios. This underlines the critical importance of accurately representing snow accumulation in Earth system models, and lays the groundwork for important future work towards more accurate representation of the cryosphere and cryospheric processes in such models.

The article is generally well written, and lays out its scientific rationale, discussion and conclusions in a clear, concise and well-structured fashion. There are relatively few grammatical and punctuation errors, but a number of ambiguous statements and uses of modeling jargon that may challenge the general reader (see SPECIFIC COMMENTS below) - but these can be easily fixed. Re the figures, these are largely informative and well presented, but I'm afraid that I'm colour blind so struggled with some of them – notably Figures 4, 6 and 7. The paper is also well referenced, with an appropriate number and quality of references.

Thank you for the positive statements about our manuscript.

We have updated the colors used in Figs 3-7 and Figs 12, 13 to use a more color-blind friendly template.

I have a number of questions/issues regarding the methods used and the results and their interpretation – that I feel need addressing before the paper can be published. First up, it's not clear how important snow properties (apart from thickness) – such as density, thermal conductivity and albedo - are treated/parameterised in the model, if at all. Other issues relate to the treatment of wind-blown snow removal/redistribution, and the conversion of snow to snow ice – the Methods section needs more information on the how snow is treated/parameterised in the model. Also, current observations of the thickness distributions of Antarctic sea ice and its snow cover are inadequate to validate model results such as those shown here e.g., in Figure 1d-e. A further question relates to the use of 70-90 deg as the geographical domain for both the northern and southern hemispheres i.e., in Figure 7. I can understand this for the Arctic, but not for the Antarctic – where 70-90 deg S mainly covers the Antarctic ice sheet and only partially covers the Antarctic sea-ice zone. There's also an apparent discrepancy between the contributions of to the mass balance in the model output (low) versus that which is typically observed (high). These and other questions/issues are outlined in SPECIFIC COMMENTS below.

As discussed further below, we now include more information in the methods section on how snow on sea ice is treated within the model. We also discuss how model limitations may affect the results. This

includes more discussion of the simulated sea ice mass budget and possible discrepancies with observations.

It is a good point about the geographical domain used for the atmospheric analysis in Figure 7. We now analyze the conditions for a more appropriate sea ice region in the Antarctic (60-80S) and find that the snowfall amounts do not decline with greater snow volumes. A revised figure 7 and associated text are now included in the manuscript.

Moreover, the Conclusions section would benefit from discussion of possible caveats and limitations in this study, and also future work that could or needs to be carried out in both observation and modeling. This could again highlight key gaps in our understanding of snow in the sea ice systems, the seasonal, regional and hemispheric dependence of the relationships, and the need for more large-scale observations of snow thickness and properties as well as accurate precipitation rates (see Webster et al., 2018). It should also address the snowfall v accumulation discrepancy factor (due to wind-blown snow redistribution/loss).

We have added an additional paragraph in the conclusions regarding caveats and limitations of the study and future work that is needed.

Given the importance of this work, the substantial large-scale conclusions reached and the fact that it represents substantial progress beyond current scientific understanding, may I recommend publication pending the authors addressing both the minor and more major comments/issues. In all cases, I hope that my comments/suggestions help.

Thank you so much for your careful read of our manuscript and the many helpful suggestions. We respond to these below.

**2. SPECIFIC COMMENTS**

2.1 The Introduction puts the study and its importance nicely into context, including previous work in the field. Possibly one thing lacking is equal coverage in the Introduction of snow on Arctic and Antarctic sea ice, and important properties and attributes therein from the perspective of this study. Notably, the 3 paragraphs from lines 40 to 69 almost exclusively focus on the Arctic. Please provide more information on what is known – and what is not known – about Antarctic snow on sea ice, and how it is relevant to this study.

We have added an additional paragraph to the introduction regarding snow on Antarctic sea ice and its relevance to this study.

2.2 Line 21 – what are these competing processes and feedbacks?

We clarify that these are competing processes and feedbacks that affect the melt and growth of sea ice (as are further articulated earlier in the paragraph).

2.3 Line 34 – the assumption here is that sea ice thickens by thermodynamic growth alone, whereas dynamics are also important – this needs stating/clarifying here.

We now clarify that ridging is included in the model (within the Methods Section) and also discuss the role that dynamics plays in transporting sea ice and causing ridging (in the paragraph where the ice mass

budget terms are discussed). We do clarify that these dynamic factors do not create or remove ice but instead redistribute it spatially or within the thickness distribution in a gridcell.

2.4 Line 36 – high snow albedo also reduces solar heating of the underlying ocean.

We have added "underlying ocean" to this sentence.

2.5 Line 48 – needs Andreas and Ackley reference.

Reference has been added.

2.6 I didn't fully understand Lines 53-57.

We have added a final clarifying sentence to this paragraph as follows: "Thus previous work suggests that for the evolving Arctic thermodynamic sea ice mass budgets, changing snow conditions have competing influences by reducing the albedo and thereby increasing summer melt and by increasing conduction of heat through the ice and thereby increasing winter growth."

2.7 Regarding treatment of snow on Antarctic sea ice and the effects of snow on sea ice simulations - may I also suggest referring to: Wu et al. 1999 (Wu, X., W.F. Budd, V.I. Lytle, and R.A. Massom. 1999. The effect of snow on Antarctic sea ice simulations in a coupled atmosphere-sea ice model. *Climate Dynamics*, 15, 127-143).

We now discuss the results from Wu et al. in both the introduction and conclusion section of the manuscript.

2.8 Line 80 - State that SHEBA is an Arctic campaign.

We now spell out the SHEBA acronym to clarify that it is an Arctic campaign.

2.9 Lines 87-88 – as stated, another important part of this question is how snow affects the retreat and duration of sea ice coverage, and whether there is regional and hemispheric dependence.

We prefer to keep the question very high level but clarify later in the paragraph that we examine the regional and hemispheric dependence. We do not mention retreat and duration as we provide no explicit metrics in our analysis on the timing of ice retreat or the ice duration length.

2.10 While the emphasis and focus is on snow accumulation/thickness, it is not clear from the Methods Section how snow properties (apart from thickness) are treated/parameterised in the model – if at all. These properties include snow density, related thermal conductivity, and grain size as it affects albedo/light transmission etc. What values are used for the different snow parameters (including albedo)? Also – how is snow converted into snow ice? Please include more information in the Methods section on how the snow is treated, and the possible caveats/limitations.

We now provide more information on how snow on sea ice is simulated within the Methods Section. This includes the use of constant density, constant snow thermal conductivity, and a prescribed grain size for dry snow that will grow when reaching melting conditions. We also clarify the snow-ice formation parameterization and provide more information on missing processes impacting the snow including the lack of wind-blown snow redistribution or loss to leads.

2.11 By the same token, it is not clear whether Antarctic snow is parameterised differently to Arctic snow in the model. What are the differences between the physical, optical and thermal properties of snow on Arctic versus Antarctic sea ice, and do they make a difference to (the modelling of) sea ice mass balance and areal coverage? – and if so, how?

There is no difference regarding how snow is parameterized in the Arctic or Antarctic which we now clarify in the methods section.

2.12 Also, it's not clear how the model treats or accounts for any discrepancy between snowfall and snow accumulation, given horizontal redistribution of the snow by winds. Sorry if I'm missing something here, but I'm just wondering whether wind speed is taken into account in the model re the different climate state scenarios, and whether increasing (change in) wind speed can also be tested as an additional factor affecting snow accumulation. This point is stimulated by the tendency of snowfall over Antarctic sea ice to typically occur under stormy, windy conditions – with wind-blown horizontal redistribution being a major process determining the snow thickness distribution and also snow loss into leads. In other words, snowfall does not equal snow accumulation. This factor is acknowledged in lines 75-77 of the Introduction – but it's again not clear how or whether this "discrepancy factor" is accounted for here and, if not, whether this is an issue.

The model does not account for any wind-blown snow effects, including possible snow loss to leads. We now clarify that this is a missing process within the Methods Section. We also discuss this and other limitations within the Conclusions.

2.13 Also - while snowfall may increase in a warmer climate scenario, will this be compensated by increased wind-blown "loss" (including sublimation) in terms of snow accumulation on sea ice?

The model does not include wind-blown snow loss to leads but does include sublimation. We now clarify this within the Method Section and discuss model limitations and the need for future work within the Conclusions.

2.14 These additional snow factors/properties – on top of snow accumulation alone – may potentially be important in terms of their effects on sea-ice mass balance. My suggestion would be to state this in the Introduction (as possible caveats), then revisit in the Conclusions i.e., state there that the results presented here are based on accumulation only, and that more work is required (if this is the case). This would make a more convincing case for focussing on snow accumulation and thickness alone here. Maybe future work would/could involve sensitivity studies to account for what is currently known about Arctic and Antarctic snow physical and optical properties on sea ice.

Thank you for this suggestion. We now better describe the model limitations in the methods section (no wind-blown snow effects, constant snow thermal and optical properties) and provide further discussion on model limitations and discuss the implications for future work within the conclusions.

2.15 Regarding these comments, may I suggest that the Methods section focus more on snow, how it is modelled here, and why that approach is taken. By the same token, the paper would benefit from providing more details on the nuts and bolts of the sea ice, atmosphere and ocean components of the model, and how hemispheric differences are catered for – in a Supplementary Section. While relatively concise, the current description provided in Lines 107-114 could be expanded upon in a Supplementary Section, to also remove jargon and aid/enhance the reader's understanding. It could also highlight current strengths and weaknesses of the model; this information is currently lacking.

We have added text to better describe the sea ice model and more comprehensively describe how snow on sea ice is simulated. We also provide more information on model limitations, including in the sea ice mass budget parameterizations.

2.16 Lines 108-114 – not clear to a non-modelling person – jargon.

We now provide more information on the slab ocean model and what it represents.

2.17 Line 121 – what environmental conditions? Needs more detail.

There are many factors that affect precipitation in the model and a full description of this is beyond the scope of the manuscript. We do now clarify though that it depends on temperature, humidity and cloud and aerosol properties and provide a reference for a more comprehensive description.

2.18 Lines 126-128 – I read this a number of times but still didn't fully understand.

We have clarified the wording here.

2.19 Line 130 – what atmospheric feedbacks? Give details here.

We have modified this sentence and clarified that surface heat and moisture fluxes and their influence on air temperature, convection, cloud properties, among other conditions will be affected.

2.20 Lines 144-147 – these statements need backing up with references.

We have added references regarding the observed sea ice changes in the late 20th century.

2.21 Lines 148-149 – it could be argued that current knowledge of large-scale sea ice thickness and its variability and change in space and time is very poor indeed and inadequate (see IPCC SROCC report etc.).

We now note the lack of sea ice observations with a reference to the IPCC SROCC report.

2.22 Line 154-155 – needs backing up with a reference.

This line describes results from the model simulation and so we are uncertain what reference should be given. We have clarified that what is discussed here is the **simulated** snow thickness pattern.

2.23 Lines 158-161, Figure caption – make it clear here whether these are observed or modelled.

We now clarify that the figure shows simulated conditions.

2.24 In the Results, one thing that struck me about Fig 2b – the SH mass budget control run – is the relatively low contribution of frazil ice (compared to congelation ice), as discussed in Lines 182-192. This is different to what is typically observed around Antarctica, with a relatively high proportion of frazil due to the highly-dynamic and turbulent conditions there. See for example: http://aspect.antarctica.gov.au/home/about-sea-ice/ice-formation (based on Worby et al., 1998): "Analysis of 173 cores taken on six voyages into the East Antarctic pack between 1991 and 1995 revealed that on average the pack was comprised of 39% columnar ice, 47% frazil ice and 13% snow ice, with other ice types making up the remaining 1%. These figures indicate the importance of the dynamic processes within the pack, which favour the growth of frazil ice. Snow ice is also seen to make a significant contribution to the total ice mass of the region." See also Lange and Eicken 1991 - (Lange, M., & Eicken, H. (1991). Textural characteristics of sea ice and the major mechanisms of ice growth in the Weddell Sea. *Annals of Glaciology, 15*, 210-215. doi:10.3189/1991AoG15-1-210-215). I'm not quite sure what to suggest here, given the discrepancy between this general understanding of the composition of Antarctic sea ice and the model results vis a vis the contribution of frazil ice.

We now provide information on some of the limitations in the sea ice mass budget parameterizations for lateral melting and frazil ice production and cite studies which have compared the model used here with models with a better lateral melting representation and more sophisticated frazil ice formation parameterization. We also note that these studies show higher lateral melting and frazil ice formation with the improved model parameterizations.

We have also added a sentence regarding the model discrepancy with observations when discussing the Southern Hemisphere ice mass budgets and provide a reference to Worby et al, 1998 and Lange and Eicken, 1991.

2.25 Line 202 – I didn't understand this sentence.

We have clarified this sentence.

2.26 Line 231 – what exactly are these different factors? Also, this is rather ambiguous, as different factors control ice edge location in different regions – at least around Antarctica.

We have removed this statement since the correlation of Arctic ice thickness and summer ice area (as shown in Blanchard-Wrigglesworth et al., 2011) is more relevant.

2.27 Line 234 – what is meant by "reduced annual cycle"?

We clarify here that this refers to the annual cycle in ice area.

2.28 Lines 294-296 – these findings are similar to those of Wu et al. 1999 (Wu, X., W.F. Budd, V.I. Lytle, and R.A. Massom. 1999. The effect of snow on Antarctic sea ice simulations in a coupled atmosphere-sea ice model. *Climate Dynamics*, 15, 127-143).

We now discuss results from Wu et al., 1999 in the introduction and conclusion sections of the manuscript.

2.29 Line 306 – why is lateral melt low across all Antarctic simulations? Lateral melt is thought to be a major factor in the annual meltback of Antarctic sea ice (numerous papers by Gordon etc.). Once again, it should probably be stated that ice formation/advance and retreat/meltback are driven by not only thermodynamics but also ice dynamics.

As mentioned above: "We now provide information on some of the limitations in the sea ice mass budget parameterizations for lateral melting and frazil ice production and cite studies which have compared the model used here with models with a better lateral melting representation and more sophisticated frazil ice formation parameterization. We note that these studies show higher lateral melting and frazil ice formation with the improved model parameterizations."

We now explicitly mention the role of ice dynamics for the sea ice mass budgets in our description of the different ice mass budget terms. Specifically, we state: "Dynamic processes do not directly act as a gain or loss of ice mass but will transport sea ice regionally and drive ice ridging, which causes conversion of thinner ice to thick ice with a smaller areal coverage."

2.30 Lines 308-309 – how realistic is this assertion that most Antarctic ice forms in coastal regions? Other studies have shown that formation within the pack and at the ice edge are also very important, depending on region.

We now clarify that this result is what the model produces. We also have added information on some of the limitations on the ice mass budget parameterizations used in the model.

2.31 Line 376 – does rainfall occur over the entire sea-ice zones, and is this rainfall area alos seasonally dependent? Also, does rainfall remove an existing snow cover, or change its albedo?

Rainfall will occur according to the environmental conditions and so can occur over the sea ice if conditions are warm enough. This varies seasonally, with more rain in the summer and little rain over sea ice during winter. So basically, the rain-season duration lengthens in the 2xCO2 climate, which we now mention in the manuscript.

We now clarify in the model description section that rainfall does not directly impact the simulated snow cover optical or thermal properties.

2.32 Lines 415-421 – May I also suggest including discussion of the new results in the light of the Wu et al. (1999) paper in paragraph 1 of the Conclusions i.e., comparing findings of that study with this one. This also relates to Lines 440-447.

We now discuss results from Wu et al., 1999 in the introduction and conclusion sections of the manuscript.

2.33 Line 425 – doesn't this also depend on ice concentration? i.e., a lower concentration or area of sea ice coverage enhancing air temperatures.

We meant this statement to just be a simple statement of the Arctic atmospheric temperature changes in our sensitivity simulations without implying a direct causality to any single factor. We have revised the statement accordingly.

2.34 Lines 435-438 – I didn't quite follow this argument of higher growth rates under thicker perennial ice. Does not congelation ice growth decrease with increasing ice thickness? Also, what about rapid frazil ice formation in more turbulent conditions of the marginal ice zone? Moreover, the conceptual model proposed is again only based on consideration of thermodynamics. How does the model account for dynamic thickening by deformation?

The argument is associated with the sensitivity to snow. It states that ice in the perennial ice zone grows more when it has less snow. As snow is a very effective insulator, this seems quite straight forward. Dynamic deformation will not increase the ice mass but instead redistributes it (generally causing thinner ice to deform into thicker ice with a reduced area). This is now more clearly discussed in the model description and in the description of the different terms in the mass budget.

2.35 Line 466 – suggest changing to: "….is a "friend" to sea ice rather than a "foe" (see Sturm and Massom, 2017)…." NB This reference is also incomplete in the Reference List, missing the chapter title.

Added the reference as suggested and added chapter title to the reference list.

2.36 Page 22 – May I suggest that the Conclusions section ends with discussion of possible caveats and limitations in this study, and also future work that could or needs to be carried out in both observation and modelling. This could reiterate key gaps in our understanding of snow in the sea ice systems, the seasonal, regional and hemispheric dependence of the relationships, and the need for more large-scale observations of snow thickness and properties as well as accurate precipitation rates (see Webster et al., 2018). It should also address the snowfall v accumulation discrepancy factor (due to wind-blown snow redistribution/loss).

We now provide an additional paragraph discussing model limitations and the need for future work in both the modeling and observational context.

2.37 Page 11 – probably better to add full figure captions to Figures 5 and 6.

We have added the full figure captions.

2.38 General question - What impact does (changing) snow accumulation have on the timing of sea ice advance and retreat, and the resultant duration of annual coverage in both polar regions?
Unfortunately, we do not have the daily data from the model simulations that would be necessary to assess the timing of ice advance and retreat.

**3. TECHNICAL CORRECTIONS**

3.1 Consistency – use either snow depth or snow thickness throughout.
We have changed the wording to "snow depth" throughout.

3.2 Line 19 – grammar
Wording has been revised.

3.3 Line 24 – reference needed.
We have added a reference here.

3.4 Lines 258-259 – grammatical error.
Fixed

---

## Author Comment (AC3)

Response to Reviewer 3.

Review of "The influence of snow on sea ice as assessed from simulations of CESM2" by Marika M. Holland et al

General comments:
The impact of snow on sea ice is difficult to assess due to the relative lack of long-term observations in remote polar regions. Given the many, competing feedbacks between snow on sea ice and sea ice itself, it is important to quantify these feedbacks and their climate impacts. In my view, this paper significantly adds to our current understanding of snow on sea ice and its impacts. It presents a study of the influence of snow thickness on sea ice in both pre-industrial and 2xCO2 experiments in coupled slab-ocean CESM2, varying the snow thickness by multiplying the snowfall on sea ice by different constant factors. The paper finds that overall, in both hemispheres, snow on sea ice tends to result in increased sea ice volume and cooler temperatures, although the ice mass budget response differs between the hemispheres. In a 2xCO2 climate, the study finds that Arctic sea ice sensitivity to snow depth is reduced, whereas the Antarctic sensitivity is similar to the pre-industrial climate.

The paper is overall well-structured, clear, and well-written, and presents results that, in my view, are of high scientific interest. It is well-referenced, with an appropriate number of references. There is a good number of figures, and the figures are generally clear, although I have a few minor suggestions below. There are some parts of the paper where I think some additional detail would be helpful, (as explained in my comments,) but overall, I strongly recommend that this paper be accepted for publication, with minor revisions.

Thank you for your helpful comments.

Specific comments:

My first suggestion pertains to the use of the slab-ocean model (SOM); I recommend that the authors discuss some of the associated caveats in more detail in the paper. (To be clear, I do not think the use of a SOM is inappropriate for this study; I simply would encourage the authors to expand some of the discussion of the SOM.) The paper mentions that the SOM used has a prescribed ocean heat flux convergence from the CESM2 preindustrial control (line 113-114). However, it has been found that ocean heat convergence weakens near the ice edge with increased CO2 (as mentioned in Bitz et al 2005, which this paper cites) and thus, I wonder if the Qflux used in this configuration may still retain some bias related to this. Also, the lack of representation of ocean dynamics in the SOM is briefly mentioned, but I think it would be helpful to briefly discuss possible impacts the lack of ocean dynamics could have on the results of this study, where applicable (eg. possibly in terms of hemispheric differences being modified by ocean dynamics).

We have added some text in the Methods Section to better describe the SOM, the implications for neglecting dynamic ocean feedbacks, and the equilibrated nature of the 2xCO2 simulations. We also return to the implications of using a SOM in the discussion of the mass budgets and in the conclusions.

Secondly, although several snow processes are mentioned, it is not always clear which of these processes are represented in the model (eg. wind-driven blowing snow, rain-on-snow, snow density and thermal conductivity, etc.). I think it would be helpful to include a brief description of which snow processes and properties are described by the model, and possibly what biases could be present from the exclusion of certain processes. (It seems to me that this is briefly discussed in places, but I think more detail would be beneficial.)

Thank you for this suggestion. We now more clearly describe how snow on sea ice is modeled within the methods section. In the conclusion section, we return to some of the limitations in the snow model representation and the need for further work to understand snow processes and to improve the model representation.

Finally, I think there could be additional discussion included in the conclusion about what uncertainties remain, and possible next steps.

Within the conclusion section, we now include an additional paragraph that discusses model limitations and what this suggests for future work.

Specific comments line-by-line:
(N.b.: These comments may be taken as suggestions .)

105: Is this due to processes not being represented in the model?
It is not entirely clear why the snow is thin and accumulates too slowly on Arctic sea ice in CESM2. We expect that it may be due to the ice being too thin and a too warm climate. We choose not to speculate here though since the reasons are uncertain.

116: Can you be more specific about what "mostly equilibrated" means in this context?
We now provide more clarification here.

117: Perhaps describe how missing ocean dynamical feedbacks could impact the results here.
We now mention that differences in ice melt/growth rates across the simulations will not influence ocean mixing and heat transport. We also clarify that for the 2XCO2 simulations, the use of a SOM means that we are assessing an equilibrium climate response.

120: It would be helpful to include some description of the snow processes in the model in or before this paragraph (or somewhere else in the section, where appropriate).
We now include an extra paragraph that describes the snow processes in the model.

145-146: There seems to be a missing citation here for "lack of significant anthropogenic ice loss that has been observed in recent decades"
We now provide a reference.

Figure 1: If possible, I suggest that these map figures be made without the grid, since it is too faint to be visible and it seems to be producing artefacts. Also, for clarity, consider specifying the spacing of the black isolines somewhere (perhaps in the figure caption).

We have revised Figure 1 to remove the grid artifacts and include the contour interval for the black isolines in the figure caption.

Also, although snow observations are limited, I think it could be beneficial to include a comparison with snow thickness observations if those are available; perhaps as a supplementary figure if it does not fit well within the main text.
We now include a comparison of the simulated snow to observations in supplementary material.

164: Are there any ice processes not simulated by the model that could have an impact on the results?
We now mention some of the limitations in the parameterizations of some mass budget terms within the model, including the simple lateral melting parameterization and lack of variable floe sizes and simple frazil ice formation parameterization.

172: Some points that may be helpful to discuss briefly: how might the constant density impact the results? In what regions would we expect results to be most impacted by this density assumption?
We have added some information on the implications of the constant snow density and regions and seasons where it might be most impacted.

Figure 3,5: Consider aligning the months here to the seasons as in Fig. 2.
We have chosen not to align the month here as in Fig. 2 given that many people are used to looking at the annual cycle of these properties over a calendar year.

273: The term "basically equivalent" is vague; could you clarify what do you mean by this?
We now clarify here that equilibrium conditions occur when the long-term average melt and growth are equal.

Figure 10: Some of the overlapping bars are difficult to see; is it possible to make them narrower so that they overlap less?
We attempted to make these narrower but they still have considerable overlap and the figure is really not any clearer. Given this, we have chosen to keep the figure as is.

331 (and other places where ice motion is mentioned): It would be helpful to mention what drives the ice motion in this model.
We now mention in the description of the model within the Methods Section how ice motion is computed.

426: This is the first time in this paper that these cases are specifically referred to as regimes so it seems to come somewhat out of nowhere; consider mentioning it earlier in the paper.
We now mention these regimes earlier in the manuscript.

Technical corrections/notes:
111: I think "mixed-layer-averaged" should be hyphenated, or this could be rephrased to avoid hyphenation (the wording is somewhat ambiguous as-is)
We have revised the wording hear to more clearly describe the slab ocean model.

117: Missing comma after "excluded"
This sentence has been revised.

173: The 3 should be a superscript in kg/m3
Fixed.

212: I think "snow free" should be "snow-free"
Fixed

224: It would be clearer if this said something like "a significant jump in ice volume and area from the Fsnow=0 case to the Fsnow=0.25 case"; clarifying that the jump is between the 0 and 0.25 cases.
Thank you for the suggestion. Revised as suggested.

240: Missing word, should be "the colors are the same"
Fixed

258: Hyphen missing in "high-latitude"
Fixed

259: Missing word, should be "consistent with a reduced"
Fixed

300: As in 212, I think this should be "snow-free"
Fixed

304: Hyphen missing in "snow-covered"
Fixed

341: I think there should be a comma before "but"
Fixed

349,363: Hyphen missing in "snow-ice" (assuming that the convention being used is to hyphenate)
Fixed

379,415,422, etc.: Inconsistencies in hyphenation of "preindustrial" vs "pre-industrial"
Changed everywhere to "preindustrial"

References (general): Many entries here include italicization, but TC guidelines indicate to not italicize text in references.
Italicization removed.